



# Global 500 m seamless dataset (2000-2022) of land surface reflectance generated from MODIS products

Xiangan Liang[1], Qiang Liu[2]*, Jie Wang[2], Shuang Chen[3], Peng Gong[3,4]

[1] Ministry of Education Key Laboratory for Earth System Modeling, Department of Earth System Science, Tsinghua University,
Beijing 100084, China;
[2] Peng Cheng Laboratory, Shenzhen 518000, China;
[3] Department of Geography, The University of Hong Kong, Hong Kong, China;
[4] Institute for Climate and Carbon Neutrality and Department of Earth Sciences, The University of Hong Kong, Hong Kong, China

*Correspondence to*: Qiang Liu (liuq03@pcl.ac.cn)

**Abstract.** The Moderate Resolution Imaging Spectroradiometer (MODIS) is widely utilized for retrieving land surface reflectance to reflect plant condition, detect ecosystem phenology, monitor forest fire, and constrain terrestrial energy budget. However, the state-of-art MODIS surface reflectance products suffer from temporal and spatial gaps due to atmospheric conditions (e.g., clouds and aerosols), limiting their use in ecological, agricultural, and environmental studies. Therefore, there

is an urgent need for reconstructing spatiotemporally seamless (i.e. gap-filled) surface reflectance data from MODIS products. In general, there are two challenges in reconstructing seamless MODIS surface reflectance product. First, the intrinsic inconsistency of observations due to various sun/view geometry. Second, the prolonged missing values resulting from polar night or heavy cloud coverage, especially in monsoon season. To address these challenges, we built a framework for generating the global 500 m daily seamless data cubes (SDC500) based on MODIS surface reflectance dataset, which contains the

generation of a land cover-based *a priori* database, BRDF correction, outlier detection, gap filling, and smoothing. The first global spatiotemporally seamless land surface reflectance at 500 m resolution was produced, covering the period from 2000 to 2022. Preliminary evaluation of the dataset at 12 sites worldwide with different land cover demonstrated its robust performance. The quantitative assessment shows that the SDC500 gap-filling results have a root-mean-square error (RMSE) of 0.0496 and a Mean Absolute Error (MAE) of 0.0430. The SDC500 BRDF correction results showed a RMSE of 0.056 and

a bias of -0.0085 when compared with MODIS NBAR products, indicating the acceptable accuracy of both products. From a temporal perspective, the SDC500 eliminates abnormal fluctuations while retaining the useful localised feature of rapid disturbances. From a spatial perspective, the SDC500 shows satisfactory spatial continuity. In conclusion, the SDC500 is a well-processed global daily surface reflectance product, which can serve as the fundamental input for large-scale ecological, agricultural, environmental applications and quantitative remote sensing studies. The SDC500 is available at:

http://data.starcloud.pcl.ac.cn/resource/27 or https://doi.org/10.12436/SDC500.27.20230701 (Liang et al., 2023).



## 1 Introduction

The increasing global ecological and climatological challenges in recent years have heightened the necessity for a more quantitative understanding of the Earth's system, which demands long-term, high-frequency, and high-quality observation data at a global scale (Estoque, 2020). Optical remote sensing imagery is one of the most widely used data sources for Earth observation. However, about 60% of pixels in optical remote sensing images are unusable due to cloud cover, dust, and heavy aerosol situations (Claverie et al., 2015; Ju et al., 2012; Vermote et al., 2016), which are usually filled with invalid values or retain their cloudy values. These unusable pixels make the observations spatiotemporally incomplete, hindering their application in studies (Yang et al., 2022; Liu et al., 2021; Fang et al., 2019; Yuan et al., 2011; Liang et al., 2022). Typically, a time-consuming and labour-intensive pre-process is required to fill the spatiotemporal gaps. Therefore, a global, well-processed, spatiotemporally seamless (gap-filled) optical remote sensing observation product is needed to serve as a fundamental input for large-scale applications. For example, the inversion of LAI and FVC parameters in the Global LAnd Surface Satellite (GLASS) product requires gap-filled surface reflectance as input (Tang et al., 2013), and many aerosol estimation algorithms need gap-filled surface reflection to quantify the contribution of surface reflected radiation component (Yan et al., 2022).

Over the last two decades, the MODIS sensors on the Terra and Aqua satellites have provided high-quality global Earth observations at a spatial resolution of 250 m/500 m/1 km (Justice et al., 2002). This makes the MODIS products suitable for generating the global seamless observation data. However, although numerous MODIS reconstruction algorithms are proposed, their practices are mostly applied on NDVI (Cao et al., 2018; Chu et al., 2021; Li et al., 2023) or other vegetation parameters (Zhu et al., 2013; Cao et al., 2023; Ma et al., 2022; Wild et al., 2022; Ma and Liang, 2022; Chen et al., 2015; Xiao et al., 2016), the research efforts of reconstructing surface reflectance image series are scarce. The land surface reflectance is a fundamental optical remote sensing product, which is derived through atmosphere correction of the Top-of-Atmosphere reflectance observed by the satellite. As a direct-observed physical parameter, the land surface reflectance is widely used to reflect plant condition (Chen et al., 2019; Fensholt and Proud, 2012), detect ecosystem phenology (Gray et al., 2019; Zhang et al., 2004; Mao et al., 2012), monitor forest disturbance (Lizundia-Loiola et al., 2020), and constraint terrestrial energy budget (Wu et al., 2017; Jia et al., 2023). Therefore, there is an urgent requirement for a pre-processed seamless (gap-filled) global land surface reflectance data to serve as the primary input for the ecological, agricultural, and environmental applications, as well as quantitative remote sensing studies (Zhao et al., 2021; Yang et al., 2021; Jiang et al., 2022; Jiang et al., 2017; Liu et al., 2012).

Generally, there are two main challenges for reconstructing the seamless MODIS land surface reflectance data:

First, the intrinsic inconsistency of observations due to various sun/view geometry. In contrast to vegetation parameters such as NDVI and LAI, the land surface reflectance is affected by the surface bidirectional reflectance distribution function (BRDF). This creates an artificial variance in the surface reflectance time series as the observations of a specific pixel on different dates correspond to different sun/view angles. Although the BRDF effect can be useful in quantitative retrievals of surface albedo and canopy architecture parameters, it is typically considered noise and is preferred to be removed in most



applications. The MODIS nadir BRDF-adjusted reflectance (NBAR) dataset (MCD43A4) was generated from the BRDF

correction of the original MODIS land surface reflectance product (MOD09GA) for the purpose of vegetation monitoring and

phenological studies (Schaaf et al., 2012; Schaaf et al., 2002). However, as the BRDF inversion model of the NBAR dataset

requires the cumulation of multiple clear observations in a 16-day window, the resulting NBAR dataset still suffers from

missing data. Additionally, although the ideal surface reflectance should be corrected for atmospheric effects, the actual

MODIS surface reflectance product (MOD09GA) still suffers from the residual influence of cloud and aerosols, which bring

further discontinuity in to the observation data. Tang et al. (2013) used the temporal smoothing method to reduce the intrinsic

inconsistency of MODIS surface reflectance series and improve the cloud detection results. However, their method did not

consider the BRDF effect, which increased the uncertainly in the cloud detection and smoothing. In this paper, a land cover-

based BRDF correction is performed to normalize the sun/view geometry before smoothing out the time series.

    Second, while a large number of gap-filling methods have been proposed, the challenge of prolonged continuous missing

values resulting from heavy cloud coverage during the monsoon season or polar night has not been addressed well. It is

essential to have a robust algorithm to fill the long gaps in time series of observations. Typically, mathematical filters such as

rolling average or Savitzky-Golay (SG) filters are often employed to smooth out the time series of observations (Chen et al.,

2021) as well as fill gaps (Zhao et al., 2009). However, in many regions worldwide, there exist prolonged continuous missing

observations resulting from heavy cloud covers during the rainy season (e.g. in Amazon, Central Africa, South and Southeast

Asia) or due to polar nights (e.g. in the Antarctic and sub-Arctic region). The limited window length of rolling average or SG

filters makes them incapable of dealing with data containing long gaps (Kawala-Sterniuk et al., 2020). Other approaches using

frequency-domain filters to process remote sensing time series have also been proposed (Yang et al., 2015; Zhou et al., 2015).

For example, the penalized least-square regression based on discrete cosine transform (DCT-PLS) method has been applied to

generate continuous surface reflectance series from MODIS products (Xiao et al., 2015) as well as Sentinel-2 Multi-Spectral

Imagery (MSI) images (Yang et al., 2022). Although a frequency-domain filter can leverage the periodicity in multi-annual

data for time series reconstruction (i.e. the rainy season or polar night occurs at the same season of the year), the information

for filling these gaps still lacks, making the results questionable. Another set of algorithms fills long gaps based on *a priori*

information, which is generally derived from statistics of the parameter in ground measurement or satellite products (Liu et al.,

2017). For instance, Moody et al. (2005) first proposed the ecosystem curve fitting algorithm to construct a spatially seamless

albedo dataset based on the assumption that pixels of the same ecosystem class should exhibit similar phenological behavioral

curves. This methodology was later improved for LAI gap-filling (Fang et al., 2008) and also adopted to generate the global

seamless GLASS albedo product (Liu et al., 2013). A newer study used a similar method to fill gaps in NDVI series before

applying the SG filter (Chen et al., 2021). However, the gap-filling method based on *a priori* information has only been applied

for vegetation-related parameters. Its potential for filling gaps in surface reflectance is yet to be explored. In this paper, a land

cover-based *a priori* database is established to aid the gap filling process.

    The aim of this study is to generate the first global spatiotemporally seamless land surface reflectance at a resolution of

500 m (SDC500) based on MODIS products, covering the period from 2000 to 2022. An advanced framework was proposed



to address the two main challenges of reconstructing MODIS land surface reflectance data, including land cover-based *a priori* database establishing, BRDF correction, outlier detection, gap filling, and slide window smoothing. Quantitative validations

were conducted on the proposed gap-filling and BRDF correction methods. Furthermore, the performance of SDC500 was assessed at 12 sites worldwide with different land cover.



## 2 Materials

### 2.1 MODIS surface reflectance products MOD09GA (Version 6.1) and MYD09A1 (Version 6.1)

The MOD09GA product is a daily surface reflectance dataset with a 500 m spatial resolution obtained from the morning satellite EOS-terra (Barnes et al., 1998; Vermote et al., 2011). It contains a quality band that indicates whether the pixel is affected by snow, cloud, or cloud shadow. The MOD09GA dataset is the fundamental input data of SDC500, and cloud-contaminated pixels were removed in the pre-processing stage. The MYD09A1 dataset provides 8-day surface reflectance at a 500-m spatial resolution acquired by the afternoon satellite EOS-aqua. Each pixel of MYD09A1 contains the best observation

among the 8 days. Both the MOD09GA and MYD09A1 data were accessed from the Land Processes Distributed Active Archive Center: https://lpdaac.usgs.gov.

### 2.2 MODIS land cover product MCD12Q1 (Version 6.1)

MCD12Q1 is a MODIS land cover product that combines the observations from both EOS-terra and EOS-aqua (Sulla-Menashe and Friedl, 2018; Friedl et al., 2002). It adopts the IGBP land cover classification system, which divides the global land surface

into 16 ecosystems. In this paper, MCD12Q1 is adopted when building the *a priori* database for BRDF parameters for different land cover types.

### 2.3 MODIS BRDF parameter product MCD43A1 (Version 6.1)

The MODIS BRDF parameter product MCD43A1 is currently the most widely used and acknowledge remote sensing BRDF dataset (Schaaf et al., 2002; Schaaf et al., 2012), providing three BRDF kernel model parameters $f_{iso}$, $f_{vol}$, $f_{geo}$ with a spatial

resolution of 500 m, which can be used for BRDF correction. The MCD43A1 product is adopted for generating the parameters for BRDF correction, and it can be easily accessed and processed through the Google Earth Engine (GEE) platform.

### 2.3 MODIS nadir BRDF-adjusted reflectance product MCD43A4 (Version 6.1)

MODIS nadir BRDF-adjusted reflectance (NBAR) product MCD43A4 is generated to facilitate vegetation monitoring and phenological studies, in which the MOD09 surface reflectance data is corrected to a nadir viewing angles using images in a

16-day sliding widow (Schaaf et al., 2002; Vermote et al., 1997). However, as the inversion of the BRDF model needs the cumulation of multiple clear observations in a 16-day window, the resulting NBAR dataset suffers from missing data and limited its applications. The MODIS NBAR product (MCD43A4) is chosen as a comparison with our BRDF corrected results because it is also derived from MODIS data and also normalized to a standard sun/view geometry.



## 3 Methodology

The study aims to reconstruct the time series of MODIS observation in a pixel-based way. The daily observations acquired by morning satellite EOS-terra, i.e., the MOD09GA product, is the major input data. And the 8-day observations acquired by the afternoon satellite EOS-aqua, i.e., the MYD09A1 product, serve as supplemental input data, which is used only when the MOD09GA pixel is marked as cloud contaminated and the MYD09A1 pixel is marked as clear. The purpose of the differentiated treatment with EOS-terra and EOS-aqua is to make the result more like morning observation, thus more

compatible with high-resolution satellite data such as Landsat and Sentinel series, while a small part of afternoon clear observations can help to stabilize the algorithm in case all the morning observations were contaminated by cloud.

Fig. 1 illustrates the 3 main modules involved in this process: 1) land cover-based BRDF correction, 2) outlier detection and gap filling, and 3) sliding window smoothing. The primary challenge in this study is to account for the uncertainty that arises from BRDF effects and cloud contamination. A priori information plays a crucial role in the reconstruction algorithm

by providing knowledge of BRDF and phenology. Since both the BRDF correction and gap-filling functions depend on a priori information, we will first introduce the process to generate the a priori database.




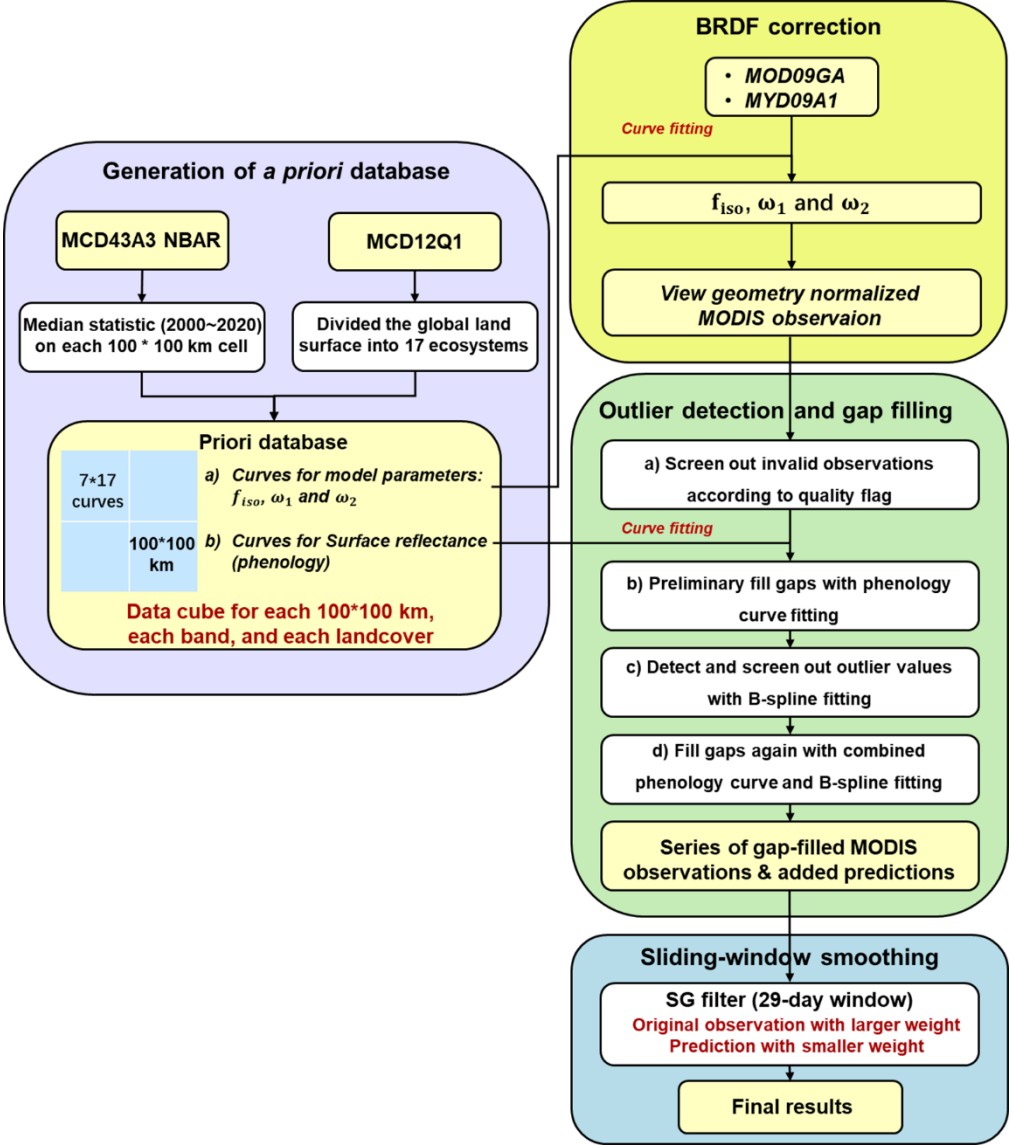

**Figure 1: The framework for global 500 m daily seamless data cubes (SDC500) generation.**

### 3.1 Generation of the *a priori* database

The MODIS BRDF product MCD43A3 is adopted to generate the *a priori* database for BRDF and phenology, as it is currently the most widely used and acknowledged remote sensing BRDR dataset. Directly using the MCD43A3 dataset in a per-pixel and per-date way is not practical due to the presence of noise and missing values. So, we derive robust statistics from the MCD43A3 dataset and utilize them for BRDF correction.

    The following formula expresses the kernel-driven RossThick-LiSparse-Reciprocal (RTLSR) model:

$$\rho(\theta_s, \theta_v, \varphi) = f_{iso} + f_{vol}K_{vol}(\theta_s, \theta_v, \varphi) + f_{geo}K_{geo}(\theta_s, \theta_v, \varphi) \quad (1)$$





Where $K_{vol}(\theta_s, \theta_v, \varphi)$, $K_{geo}(\theta_s, \theta_v, \varphi)$ are two kernel functions representing volumetric scattering and geometric-optical shadowing effects respectively; $f_{iso}, f_{vol}, f_{geo}$ are the three model parameters, which are the main content in the MCD43A3 dataset.

Before doing the statistics, we apply a simple transform:

$$\omega_1 = f_{vol}/f_{iso}, \quad \omega_2 = f_{geo}/f_{iso} \quad (2)$$


where $\omega_1$ and $\omega_2$ are called BRDF shape factors. They jointly decide the shape of the angular variation of the BRDF model which will be applied to the BRDF correction in the next section. They are roughly independent to $f_{iso}$ which represents the variation of surface reflectance in spectral, spatial and phenology.

Land surface classification is also considered in building the a priori database. The MODIS land cover type product

MCD12Q1 is adopted for this purpose, and the IGBP land cover classification system within the MCD12Q1 product is adopted, which divided the global land surface into 16 ecosystems.

The cloud computing platform of Google Earth Engine (GEE) has enabled us to efficiently access and process the large global MCD43A3 and MCD12Q1 datasets. The statistics are derived by computing the median value of $f_{iso}$, $\omega_1$ and $\omega_2$ for each MODIS land band, each 4 Julian days, each land cover type and each spatial cell of 100 km*100 km. The time scope is from

2000 to 2021, and the spatial extent includes all land surfaces as well as shallow seas covered by the MCD43A3 dataset. The median statistics are chosen for their resistance to noise, and the coarse spatial resolution ensures an ample sample size of MCD43A3 pixels. However, not all land cover types have valid median statistics in each spatiotemporal cell. For example, a cell in the tropical desert may not have any snow, cropland, or forest samples. In these cases, we filled the missing statistics with the average of neighbouring valid statistics and iterated the filling process until all spatiotemporal cells contained statistics

for all land cover types. This process enabled us to generate a global a priori database for BRDF and phenology. The GEE code can be accessed at: https://code.earthengine.google.com/363b4d94090048f9e28103ad3efebfdf.

## 3.2 Land cover-based BRDF correction

To utilize the *a priori* information of BRDF and phenology, the first step is to select one sequence of time series of $f_{iso}$, $\omega_1$ and $\omega_2$ from the *a priori* database. For each input MODIS pixel, we use the coordinate of the 500 m pixel in the MODIS

projection grid to locate the nearest time series of 100km*100km statistics, which corresponds to the 17 land cover types. These statistics of $f_{iso}$, $\omega_1$ and $\omega_2$ are used to simulate 17 sequences of time series according to the actual observation date and sun/view geometry, corresponding to each of the land cover types. We then compute the correlation coefficient between the actual MODIS observation and the 17 simulated sequences of observations. The simulated series corresponding to the maximum correlation coefficient is selected as our *a priori* information.

The process of BRDF correction is also known as BRDF normalization. As MODIS is a wide-angle sensor, the sun/view angle of a pixel varies from day to day, resulting in fluctuations in the observed time series of the surface reflectance. The objective of BRDF normalization is to simulate a time series that appears as if it was taken from standardized sun/view angles.



In this approach, the standardized sun/view angles are defined as nadir-viewing and solar illuminating at 10:30 am local time, which is denoted as $(\theta_{s0}, \theta_{v0}, \varphi_0)$ . Then, the normalization is performed with the following formula:

$$\rho(\theta_s, \theta_v, \varphi) = \rho(\theta_s, \theta_v, \varphi) \frac{1+\omega_1 K_{vol}(\theta_{s0}, \theta_{v0}, \varphi_0)+\omega_2 K_{geo}(\theta_{s0}, \theta_{v0}, \varphi_0)}{1+\omega_1 K_{vol}(\theta_s, \theta_v, \varphi)+\omega_2 K_{geo}(\theta_s, \theta_v, \varphi)} \quad (3)$$

In which the $\omega_1$ and $\omega_1$ are derived from the *a priori* information.

### 3.3 Outlier detection and gap filling

In this study, the gap refers to both the unobserved area of land surface in the daily MODIS acquisition and the area blocked by cloud/heavy aerosols. These gaps need to be filled with a prediction model to build the seamless dataset. Pure mathematical algorithms, such as linear or nonlinear interpolation, can be efficient prediction models when good quality valid samples are available near the gap. However, filling the prolonged continuous gaps and gaps with high levels of noise existing in nearby valid samples can be challenging. In these cases, the ecosystem curve fitting method is the most robust gap-filling algorithm because it utilized the *a priori* information of phenology. Therefore, we have designed a strategy that combines the mathematical and the phenology curve fitting algorithms, as illustrated in the second box of Fig. 1.

Step a) first excludes the invalid observations according to the quality flag in MOD09GA. Then, step b) provides preliminary fill values with phenology fitting to increase the stability and error resistance of the gap-filling process. As the phenology fitting is more stable and less flexible than the B-spline fitting, we apply it to provide preliminary fill values which will be utilized in the B-spline fitting step to prevent overfitting. And in the step c) of B-spline fitting, the observations with a fitting absolute error larger than 2.5 times of mean are marked as outliers, which usually means undetected cloud contamination or abnormal change of surface state such as ephemeral snow. These outlier values severely destabilize fitting process and increase error. Therefore, in this study, we remove the outliers from the observation series together with invalid values indicated by the quality flag in the MOD09GA dataset.

The phenology fitting of step b) can be expressed as the optimization of the following expression:

$$y_1(t) = a_0 + a_1 F_{phe}(t) \quad (4)$$

Where $F_{phe}(t)$ is the *a priori* phenology curve, which is actually the series of $f_{iso}$ selected in section 2.2; $y_1(t)$ is the preliminary prediction values for the time series; $a_0$ and $a_1$ are parameters to be optimized by least square principle, which is minimizing the square error between predictions and valid observations in the time series.

The B-spline of step c) can be expressed as the optimization of the following expression:

$$y_2(t) = a_0 + a_1 B_1(t) + a_2 B_2(t) + \cdots + a_6 B_6(t) \quad (5)$$

Where $y_2(t)$ is the prediction by spline; $B_1(t), \cdots, B_6(t)$ are six B-spline base functions; $a_0, \cdots, a_6$ are seven parameters to be optimized by least square principle. In this step, the preliminary prediction values are included in the samples together with the valid observations, and the outliers are detected and removed in an iterative way.

In the last step, the combine-spline and phenology curve fitting of step d) is expressed as:

$$y_3(t) = a_0 + a_1 B_1(t) + a_2 B_2(t) + \cdots + a_6 B_6(t) + a_7 F_{phe}(t) \quad (6)$$

In this step, the prediction values by spline from step c) is included in the samples, and there is no outlier detection iteration. The six B-spline base functions, as well as a demonstration phenology curve, are illustrated in Fig. 2. The demonstration phenology curve is in red band (MODIS band 1) corresponding to a cropland pixel in the North China Plain.

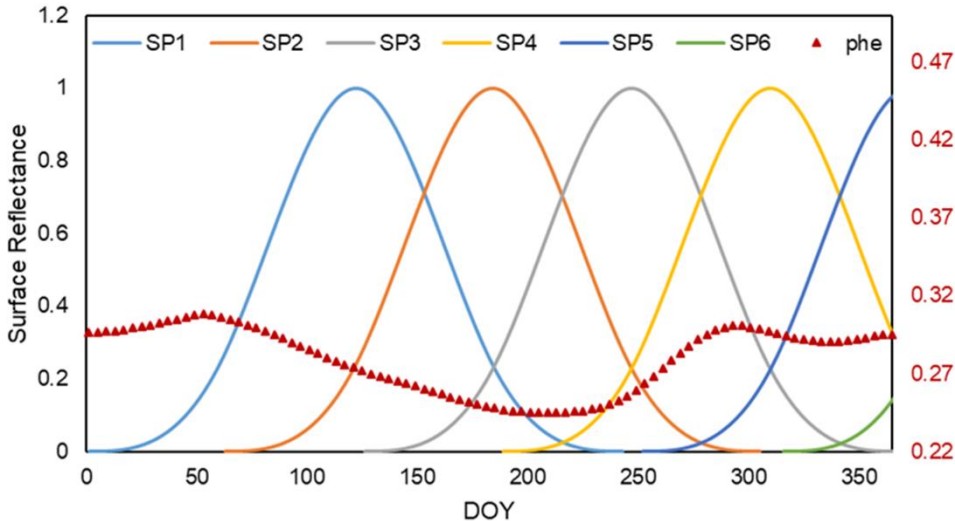

**Figure 2: Illustration of the B-spline base functions (SP1, ..., SP6) and the surface reflectance phenology curve (PHE) of MODIS**
**band 1.**



### 3.4 Sliding window smoothing

The gap-filling process produces a time series that includes both actual MODIS observations and predicted filling values.
However, this series is not completely smooth due to noise in the observations and inconsistency between the two groups. To
ensure that the dataset is smooth, we applied the SG filter algorithm (Chen et al., 2004) to the gap-filled series. This algorithm
applies a filter window of 29 days in width, which is centred on each output date and slides through the time series. The
algorithm also gives larger weight to the actual MODIS observations than the predicted filling values to ensure the resulting
time series remains as close to the original observations as possible.

### 3.5 Consideration of snow

The time series processing described above is based on the assumption that surface reflectance changes gradually over time,
which is not always the case for natural land surfaces. Abrupt changes, such as snowfall, can result in an abrupt change in
surface reflectance, and the melting of snow can also cause rapid changes that cannot be simulated with B-spline and phenology
curve. Furthermore, the BRDF feature of snow is completely different from that of snow-free surfaces. To address these issues,
a special treatment for snow was developed.

The year is divided into two parts: the snow season and the snow-free season, based on the snow indicator in the quality
flag of the MOD09GA product. To ensure a robust segmentation, a median filter with a length of 12 days is applied to the
snow indicator series. If there are no valid observations within the range of the median filter, the resulting snow status is copied
from the nearest valid neighbour.

The BRDF correction, gap-filling, and smoothing processes are then separately applied to the snow season and snow-free
season of the time series. During the processing of snow season, only the observations marked as snow are used to build a
smooth snow reflectance curve, and vice versa. Finally, the smooth snow reflectance curve and smooth snow-free reflectance
curve are combined to create the final time series of surface reflectance. As a result, the final time series may be discontinuous
in the switching date from snow season to snow-free season.

## 4 Results

The proposed framework has been utilized to generate the SDC500, a global seamless data cube, from the MODIS land surface reflectance dataset. The SDC500 encompasses all land surfaces worldwide, including shallow sea areas, and covers the temporal period from 2000 to 2022. It exhibits a spatial resolution of 500 m and a temporal resolution of 1 day. The SDC500 comprises 8 bands, with the first 7 bands representing surface reflectance in the MODIS band 1 to band 7, and the 8[th] band serving as a quality assessment (QA) band.

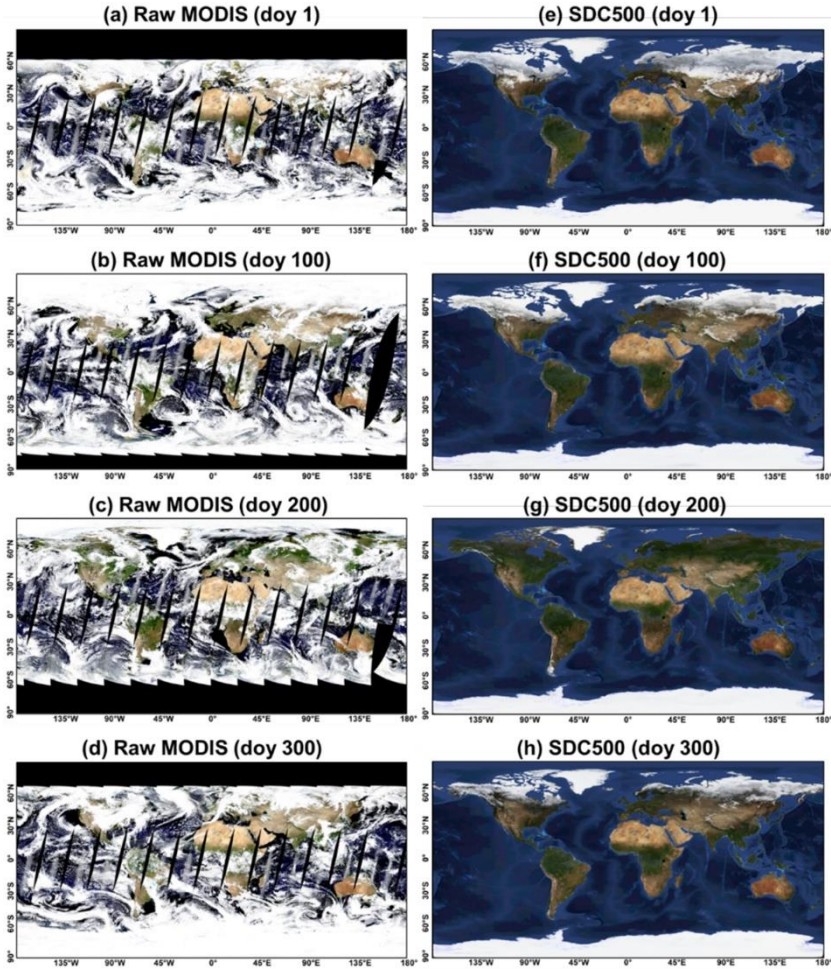

**Figure 3: True-colour composite (R: Band 1; G: Band 4; B: Band 3) imageries of Raw MODIS observation (a-d) and SDC500 (e-h) for different seasons in 2020.**

We present a joint visualization of the SDC500 and raw MOD09GA data for four specific days in 2020, resulting in real-colour composite maps at a global scale (Fig. 3). The comparison between the SDC500 and raw data highlights the accurate reconstruction of missing or cloud-covered pixel values by the SDC500. Furthermore, the SDC500 effectively suppresses

noise caused by observational conditions and corrects biased pixel values. The reconstructed SDC500 exhibits impressive spatial continuity, indicating its potential and readiness to support a wide range of applications.

To demonstrate the performance of our algorithms, we selected 12 typical sites distributed around the globe to evaluate
the quality of SDC500 (Fig. 4). Sites S1-S8 are selected to visually investigate the spatial pattern, as well as the time series profile of the reconstructed results under different climatic conditions. These sites can be grouped into 4 categories: sites S1 and S2 represent the case of tropical and subtropical monsoon climate zone, where vegetation is dense and cloud coverage presents the main challenge to remote sensing applications. Sites S3 and S4 represent the case of acrid areas where vegetation is sparse but can change significantly after occasional precipitation. Sites S5 and S6 represent the vegetated area in the
temperate climate where crop rotation and natural plant phenology is usually the focus of remote sensing applications. Sites S7 and S8 are in high latitude or high altitude, where snow covers the ground most time of the year. Thus, the surface reflectance is generally high and the snow/thaw process shapes the variation of surface status. Sites S9-S12 are selected for visualizing the result of each step of the processing algorithm in the proposed SDC500 framework. In the following section, we first visualized each step of the processing algorithm in 4.1, then, presented the image blocks of 200*200 pixels in 4.2~4.5.

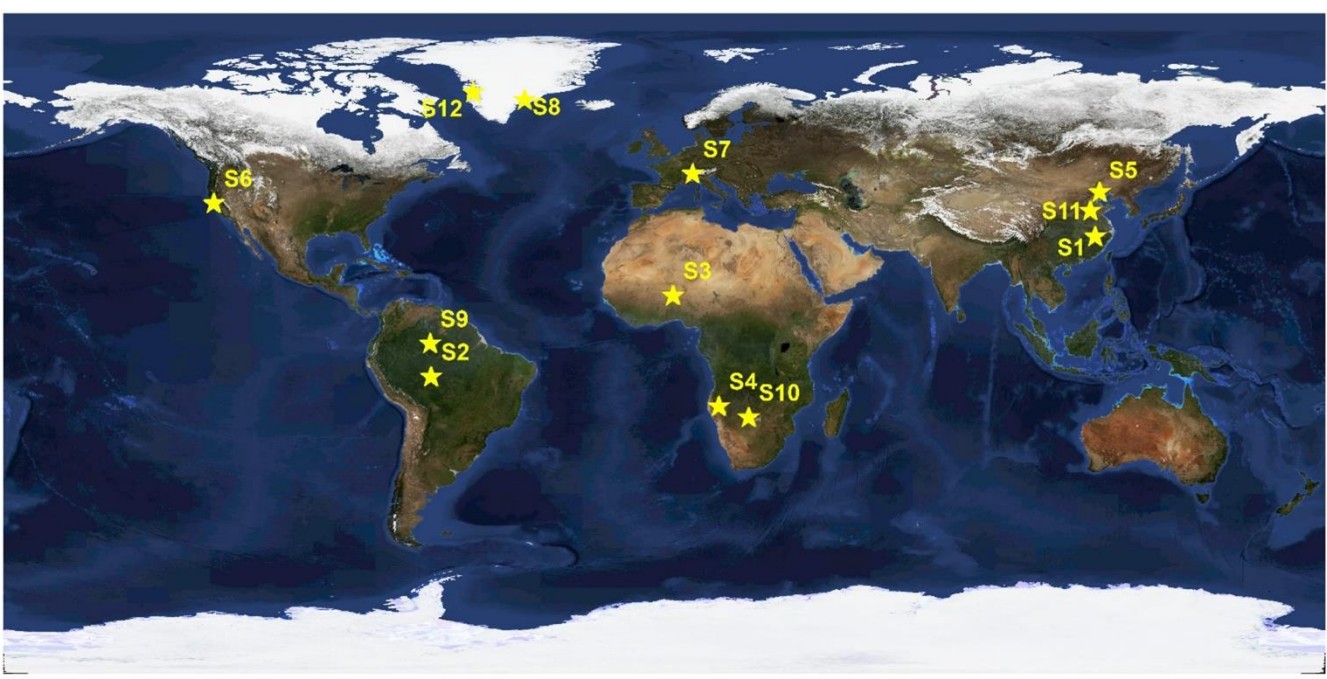

**Figure 4: Spatial distribution of the 12 validation sites.**



**Table 1: Basic information of the 12 validation sites.**

| ID | Longitude | Latitude | Tile ID | Main Land Cover | Climatic zone |
|----|-----------|----------|---------|-----------------|---------------|
| S1 | 116.5 | 29 | h28v06 | Mixed Forest | Humid subtropical climate |
| S2 | -63.5 | -8.8 | h11v09 | Evergreen Broadleaf Forest | Tropical rainforest climate |
| S3 | 2.1 | 13.4 | h18v07 | Sparsely Vegetated | Arid climate |
| S4 | 14.5 | -17.3 | h19v10 | Savana | Arid climate |
| S5 | 117.7 | 41.2 | h26v04 | Cropland | Humid continental climate |
| S6 | -122.4 | 37.8 | h08v05 | Evergreen Needleleaf Forest | Mediterranean climate |
| S7 | 7.4 | 46 | h18v04 | Evergreen Needleleaf Forest (snow) | Humid subtropical climate |
| S8 | -38.1 | 66.2 | h16v02 | Ice (Arctic) | Ice cap climate |
| S9 | -63.8 | -0.4 | h11v09 | Evergreen rain forest | Tropical rainforest climate |
| S10 | 22.6 | -19.6 | h20v10 | Grassland | Arid climate |
| S11 | 115.2 | 36.2 | h27v05 | Cropland | Humid continental climate |
| S12 | -52.1 | 67.9 | h16v02 | Ice (Arctic) | Ice cap climate |


### 4.1 Visualizing each step in the SDC500 processing

We selected temporal curves of 4 representative sites to evaluate the effect of each step in the proposed framework, respectively. Fig. 5 presents Band 2 (NIR) reflectance time series in 4 representative sites in 2010. The original MODIS sensor observations, the Outlier detection and gap-filled curve, the first smoothing results, and the final SDC500 results are shown together to
facilitate visual validation.

Site S9 (Fig. 5a) locates in Amazon Plain in Brazil, representing evergreen forest land cover, as well as a case of scarce clear observation. We can see that in Site S9, the first gap-filling result by phenology curve (green dash-dot line) is too flat, as the a priori phenology curve is in coarse resolution and lacks details. But the final SDC500 (red line) effectively reflect the seasonal dynamic while avoiding over-fitting, giving a reasonable result in the tropical zone.

Site S10 (Fig. 5b) locates in the southern part of Africa, representing grassland in the wet season and sparsely vegetated surfaces in the dry season. In Site 10, clear observations are dense and the reconstruction results are close to the unbiased average status of the surface.

Site S11 (Fig. 5c) locates in the temperate zone of the North China plain, where winter wheat is growing from October to June, and corn is growing from July to September. The discrete series (orange cross) of Site S11 show double peaks in summer
around DOY 130 and 220, and 3 valleys around DOY 50, 190, and 300. We can see that the first gap-filling (green dash-dot line) cannot approximate the double peak feature of the crop phenology, but the second gap-filling (grey line) and the final



smoothed (red line) gradually approach the discrete series (cross). It illustrates that SDC500 has stable estimation capabilities near peak and valley values.

Site12 (Fig. 5d) locates in the sub-Arctic coastal land of Greenland and is featured for the snow/thaw circle and long polar
nights in winter. In Site 12, the reconstructed result (red line) is composed of two parts: the snow-free part (140<DOY<305) is with low reflectance, and the snow-covered part is with high reflectance. In DOY range of 280~300, there are some large fluctuations in the raw observations (orange cross), a small part of the observations is marked as snow-covered and the rest are marked as snow-free, according to the raw quality flag. Our algorithm judges this period more like snow-free according to the median filter, so the final reconstructed result (red line) is closer to the lower envelope of raw observations. In the data range
before DOY 50 and after DOY 300, the site enters the polar night, and there is no valid raw observation (orange cross). The prediction in the polar night may not be as accurate as the normal date, but the reconstructed result (red line) still gives a reasonably prediction.

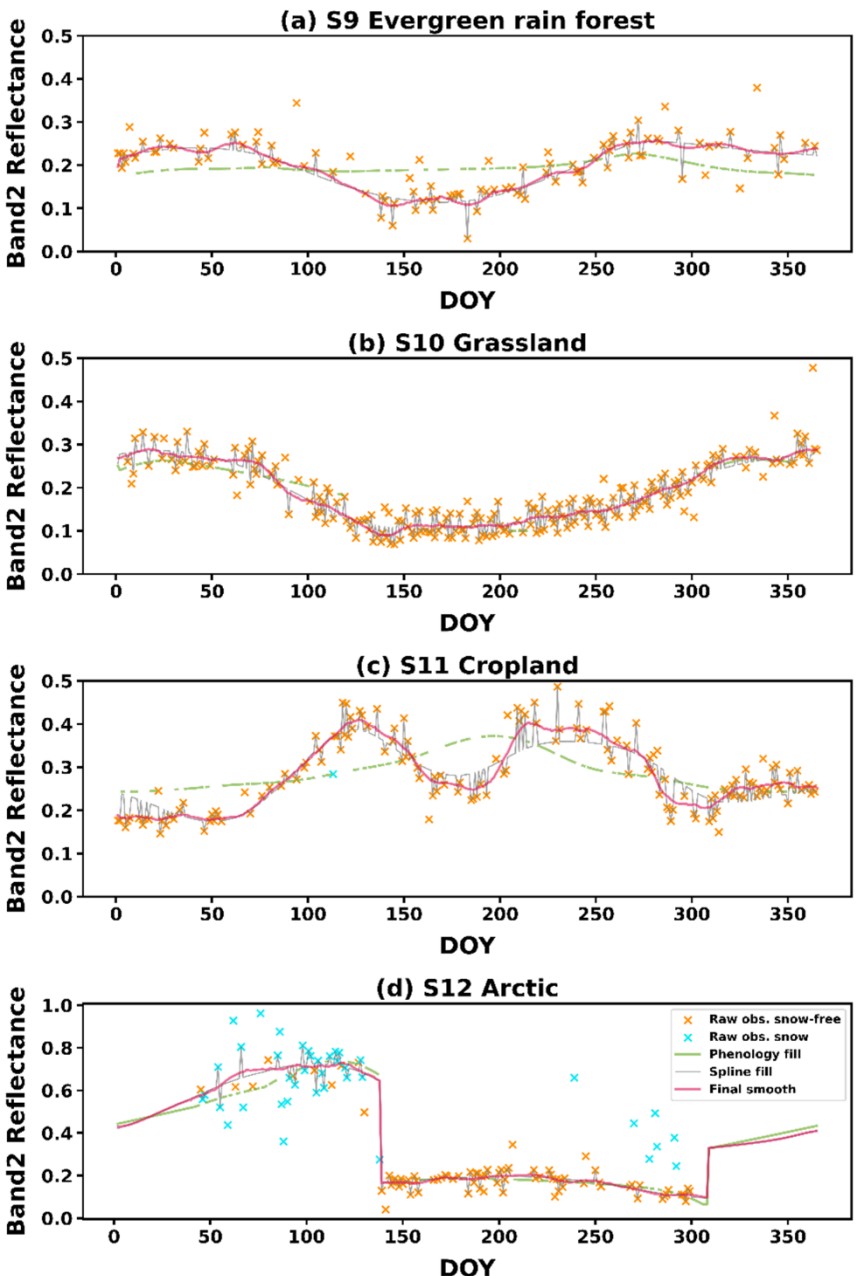

**Figure 5: Comparison of BRDF adjusted raw MODIS observations, gap-filled values with phenology curve, gap-filled values with B-spline, and the final smoothed series in typical sites: Site S9 Evergreen rain forest (a), Site S10 Grassland (b), Site S11 Cropland (c) and Site S12 Arctic (d).**

## 4.2 SDC500 in tropical and subtropical areas

In the tropical and subtropical monsoon climate zone (Fig. 6 and Fig. 7), heavy cloud cover significantly affects the satellite observations. Particularly during the monsoon season from May to August, there is continuous and dense cloud coverage,

Earth System
Science
Data

leading to prolonged gaps in valid observations. This poses a considerable challenge for seamless observation reconstruction.
Simple linear interpolation performs poorly due to the extensive missing data. But SDC500 can effectively fill these prolonged
gaps and reconstructs observation time series that closely reflect the actual conditions.

From a spatial patterns perspective, the reconstructed results of SDC500 demonstrate the preservation of effective
observation values in the central cloud-free area in site S1 on DOY 226, compared to the original observations (Fig. 6b).

Additionally, the reconstructed data in cloud-covered areas exhibit good spatial consistency with the raw MODIS data in
cloud-free areas. This spatial consistency is particularly notable considering that the SDC500 is constructed on a pixel-wise
basis. It indicates the reliability of the reconstruction algorithm.

From the temporal series perspective, despite significant cloud interference and numerous data gaps before and after this
period, SDC500 successfully reconstructs the missing data, and preserves the phenological characteristics of forests, as

illustrated in Fig. 6a. In site S2 (Fig. 7a), which is mainly covered by evergreen broadleaf forest), reflectance remains relatively
stable throughout the year, and SDC500 effectively eliminates noises and obtains stable phenological curves.

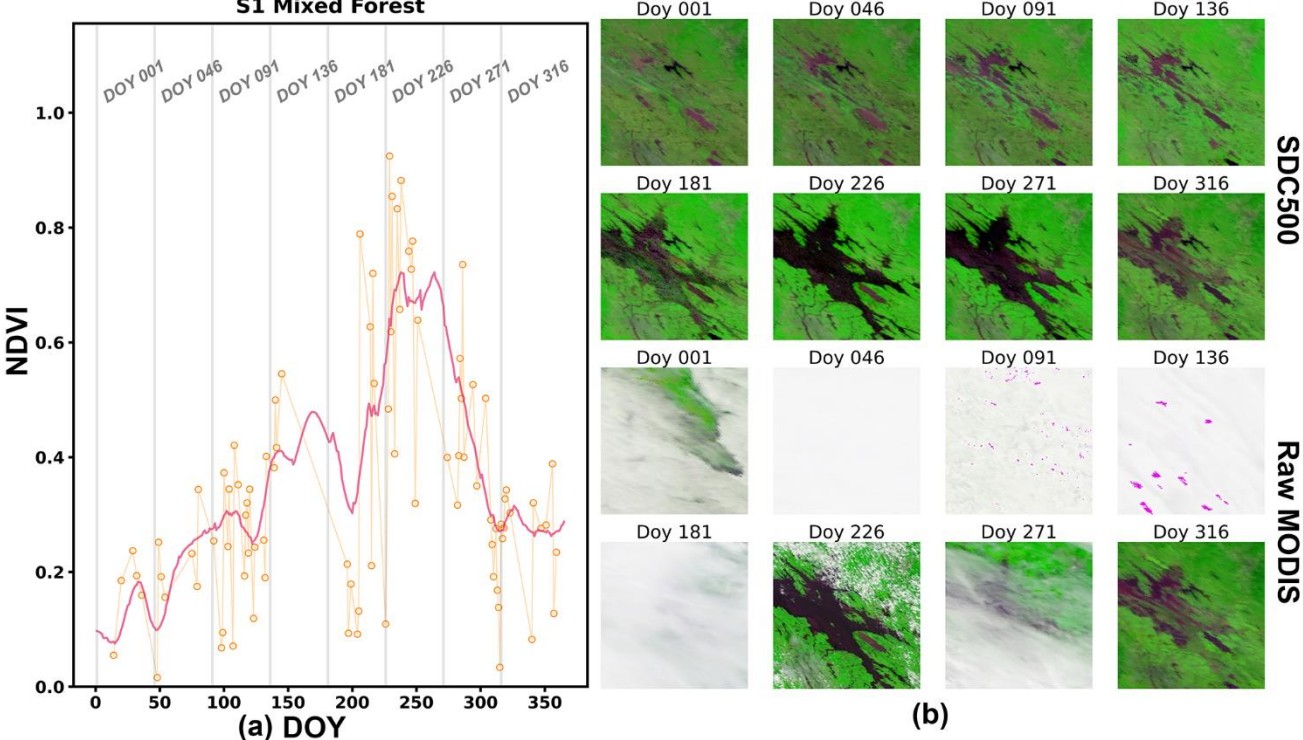

**Figure 6: Performance of SDC in site S1: (a) The NDVI curves for the central pixel, the orange points indicate the valid MODIS observation and the red line indicates the SDC500 results. (b) The spatial pattern of reconstructed and raw image blocks of 200*200**

**pixels centred around the site (R: Band 1; G: Band 2; B: Band 3).**



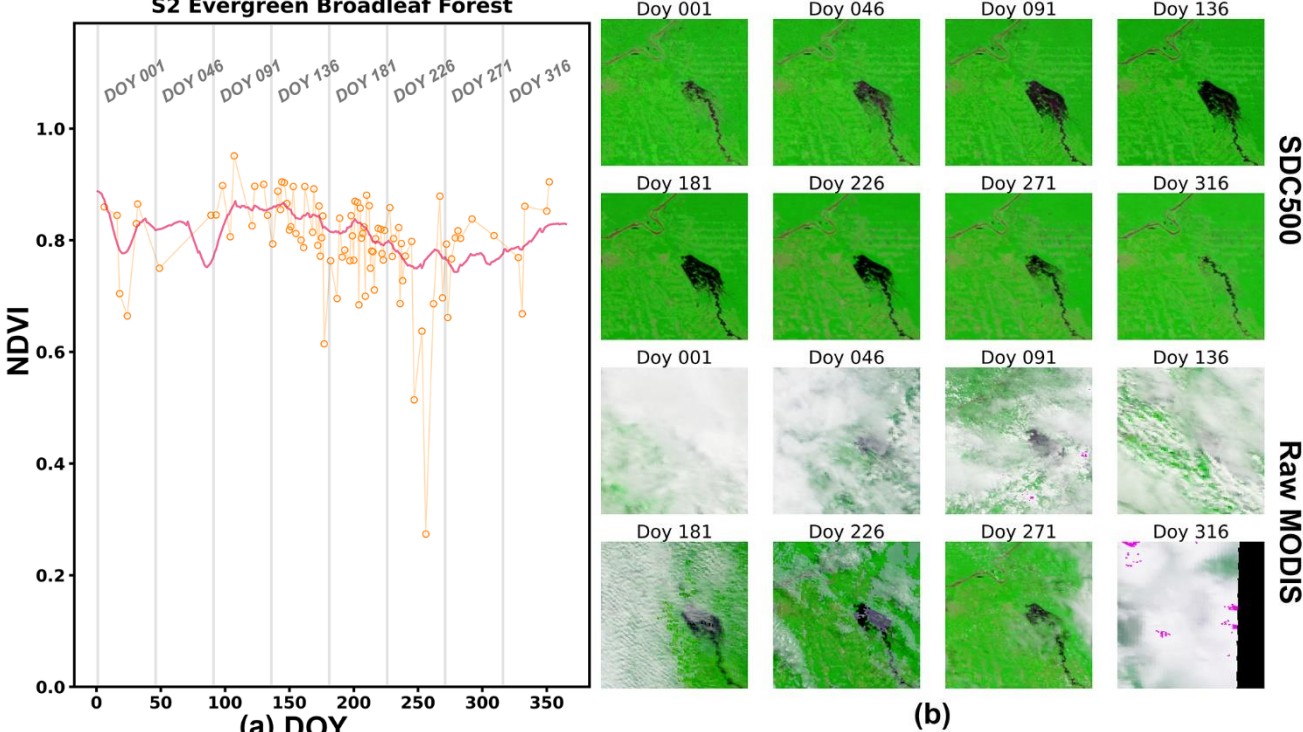

**Figure 7: Performance of SDC in site S2: (a) The NDVI curves for the central pixel, the orange points indicate the valid MODIS observation and the red line indicates the SDC500 results. (b) The spatial pattern of image blocks of 200*200 pixels centred around the site (R: Band 1; G: Band 2; B: Band 3).**

### 4.3 SDC500 in acrid areas

The SDC500 demonstrates excellent noise reduction performance in the acrid area (Fig. 8 and Fig. 9). Fig. 8b illustrates that
the raw MODIS observation on DOY 091 is affected by its large view angle, resulting in no data in the main part of the block,
and the remaining upper left corner is brighter than other raw images. However, these are effectively corrected in the SDC500
reconstructed results, which show consistency with observations on other days. The improved observation consistency on
different dates mitigates the inherent data inconsistency arising from sensor component fluctuations, observation/view
geometry, and other factors. Notably in Fig. 9b, the reflectance of water bodies in the images remains consistent throughout
the year, without any noticeable darkening or lightening, aligning closely with the physical reality. From the temporal
perspective, the NDVI series in S3 (Fig. 8b) peaks at about DOY 250 while the NDVI series in S4 (Fig. 9b) peaks at about
DOY 50, which is consistent with the fact that the two sites are in different hemispheres. And the influence of outlier values
has been satisfactorily removed, such as around DOY 200 and 250 in Fig. 8a.

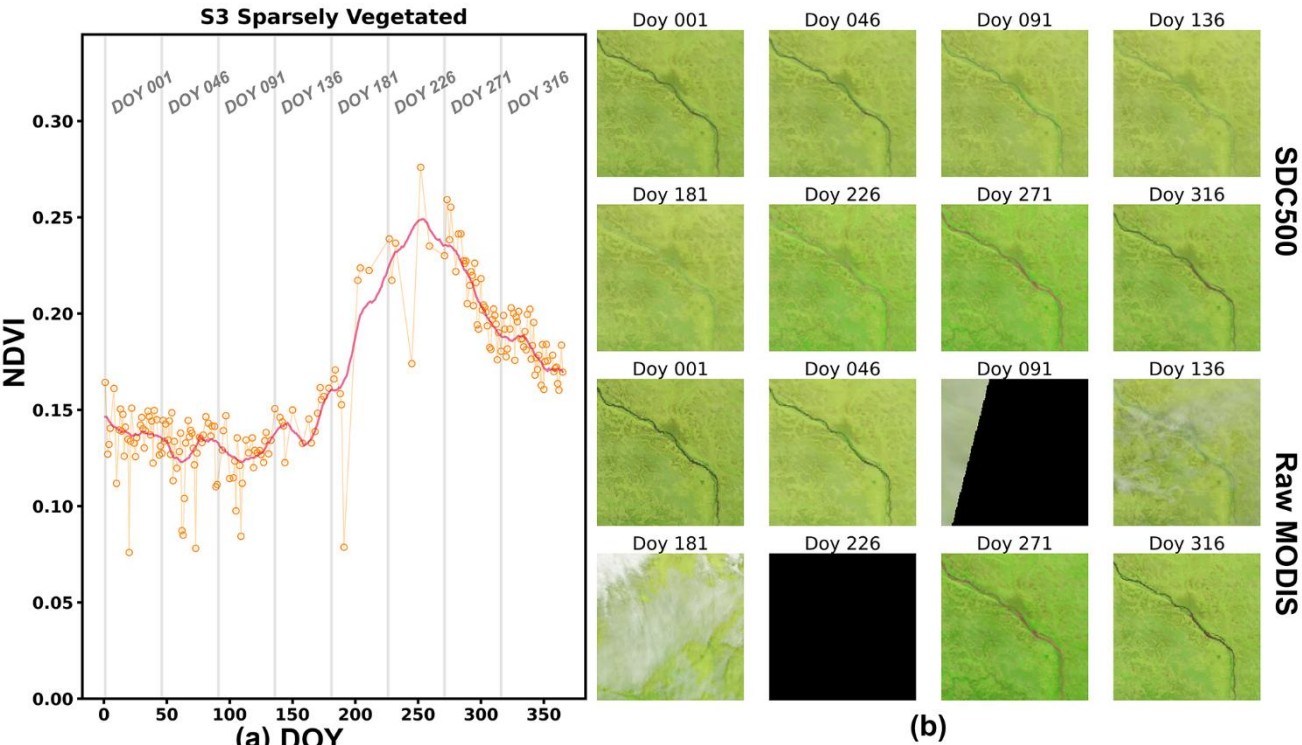

**Figure 8: Performance of SDC in site S3: (a) The NDVI curves for the central pixel, the orange points indicate the valid MODIS
observation and the red line indicates the SDC500 results. (b) The spatial pattern of image blocks of 200*200 pixels centred around
the site (R: Band 1; G: Band 2; B: Band 3).**

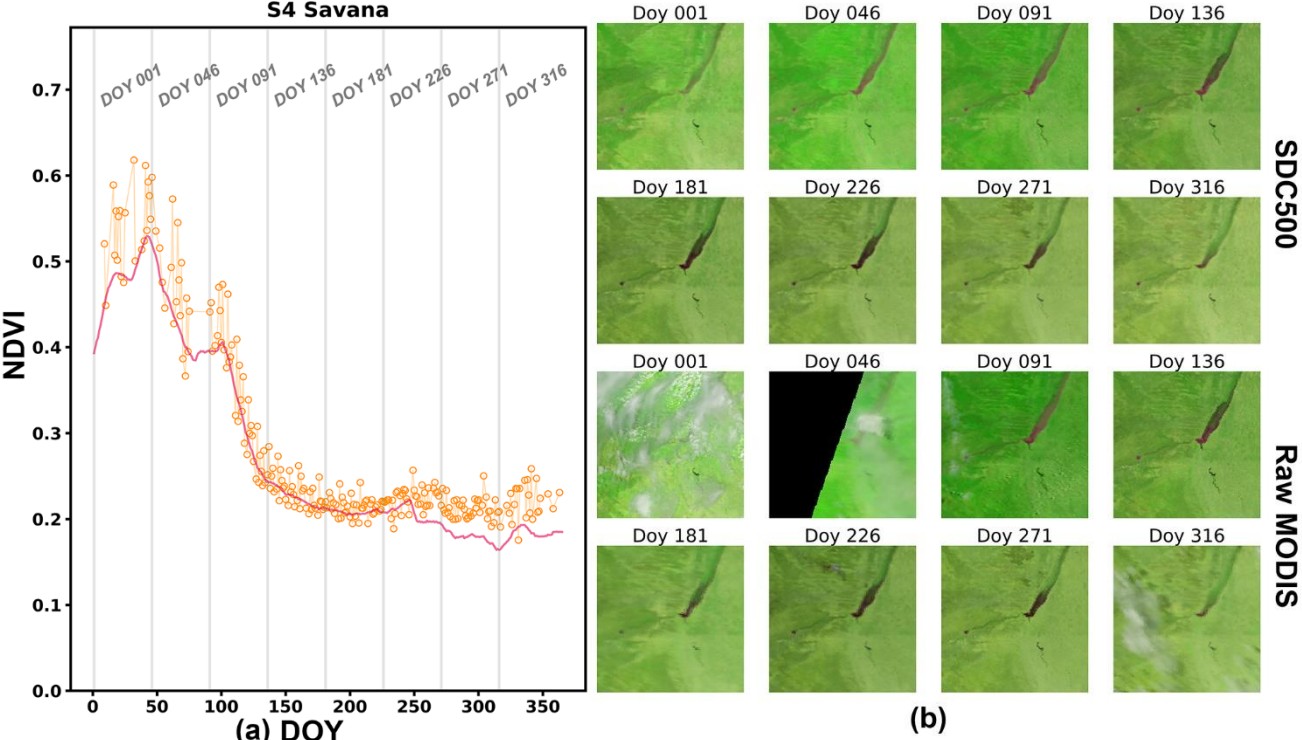

**Figure 9: Performance of SDC in site S4: (a) The NDVI curves for the central pixel, the orange points indicate the valid MODIS observation and the red line indicates the SDC500 results. (b) The spatial pattern of image blocks of 200*200 pixels centred around the site (R: Band 1; G: Band 2; B: Band 3).**

### 4.4 SDC500 in temperate climate areas

The primary land cover type of site S5 (Fig. 10) is cropland, which thrives in summer and harvests in autumn each year. Hence, the typical NDVI phenology of cropland can be used as a reference to assess the performance of SDC500. In this evaluation, we assume that the NDVI phenology in neighbouring years are similar for most of the pixels in the image block. Therefore, comparing the NDVI patterns over two years allows us to gauge the credibility of the reconstructed image. The comparison of SDC500 and raw NDVI images for the same day in 2020 and 2021 is presented in Fig. 11 and Fig. 12. The reconstructed NDVI series in Fig. 11 exhibits a transition from low to high and then back to low in the DOY range 110~310 in 2020, which aligns with the temporal curve pattern observed in 2021. Consequently, the NDVI values for the same day in different years are comparable. For example, DOY 110 in 2021 (2nd line in Fig. 13) is relatively cloud-free and can be considered as reliable reference data. The reconstructed NDVI image for DOY 110 in 2020 (3rd line in Fig. 12) agrees well with that of DOY 110 in 2021, indicating the accuracy of the reconstructed result. Similarly, DOY 260 in 2021 (2nd line in Fig. 13) can serve as reference data to validate the reconstructed result of DOY 260 in 2020 (3rd line in Fig. 12). Furthermore, certain parts of the region on DOY 160/310 in 2020 (2nd line in Fig. 12) are not affected by clouds and can be used as reference data. The reconstructed

results on DOY 160/310 in 2021 (3rd line in Fig. 13) effectively preserve the high-quality observations in these regions, exhibiting good spatial continuity with the reconstructed results in cloud-contaminated regions. As the SDC500 is produced on a per-pixel basis, the consistency between clear and cloud-contaminated pixels demonstrates the robustness of the algorithm. Additionally, the results reveal the successful restoration of low NDVI values caused by clouds, while preserving the low

NDVI values of bare land roads. This indicates the successful reconstruction of spatiotemporally continuous missing pixels by the proposed algorithm.

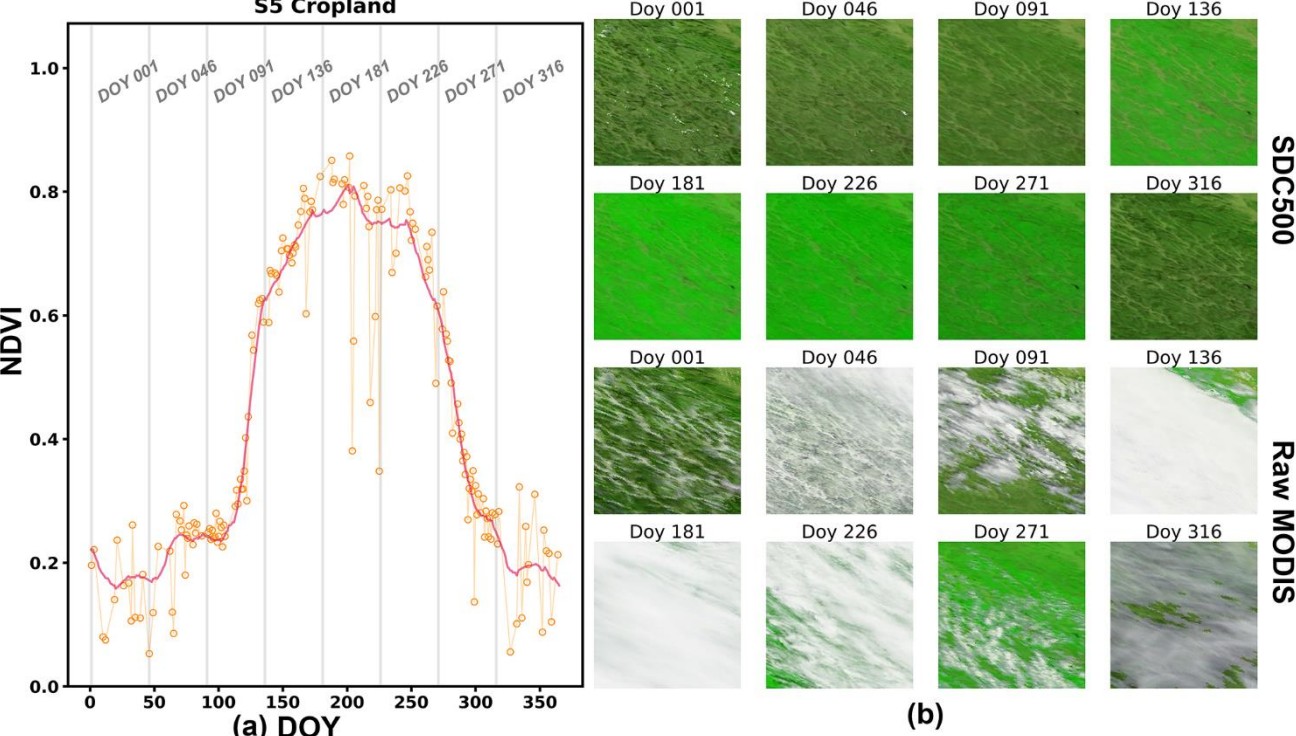

**Figure 10: Performance of SDC in site S5: (a) The NDVI curves for the central pixel, the orange points indicate the valid MODIS observation and the red line indicates the SDC500 results. (b) The spatial pattern of image blocks of 200\*200 pixels centred around**

**the site (R: Band 1; G: Band 2; B: Band 3).**

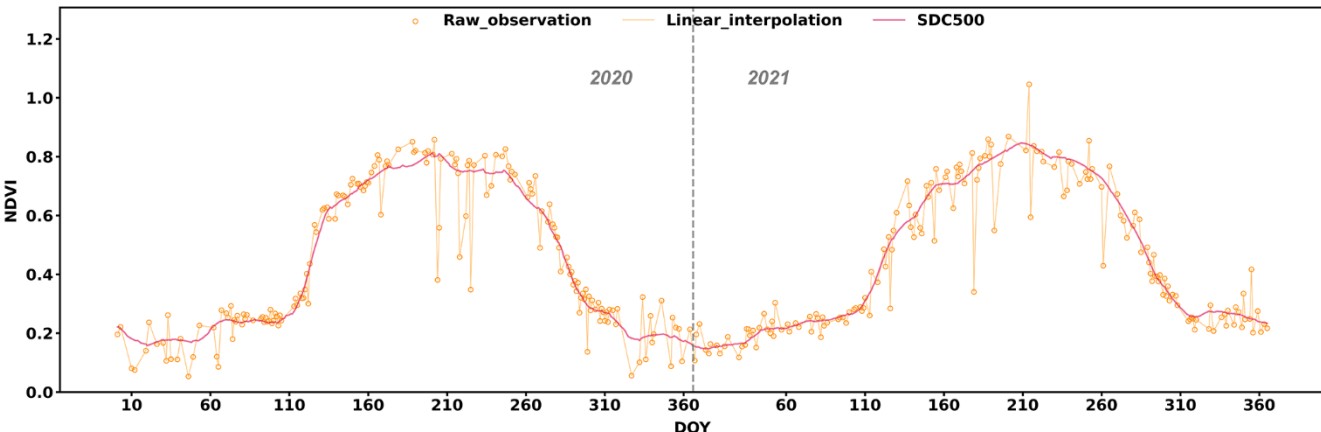

**Figure 11: NDVI curves of the central pixel in a cropland area (site S5) in the period of 2020 and 2021.**

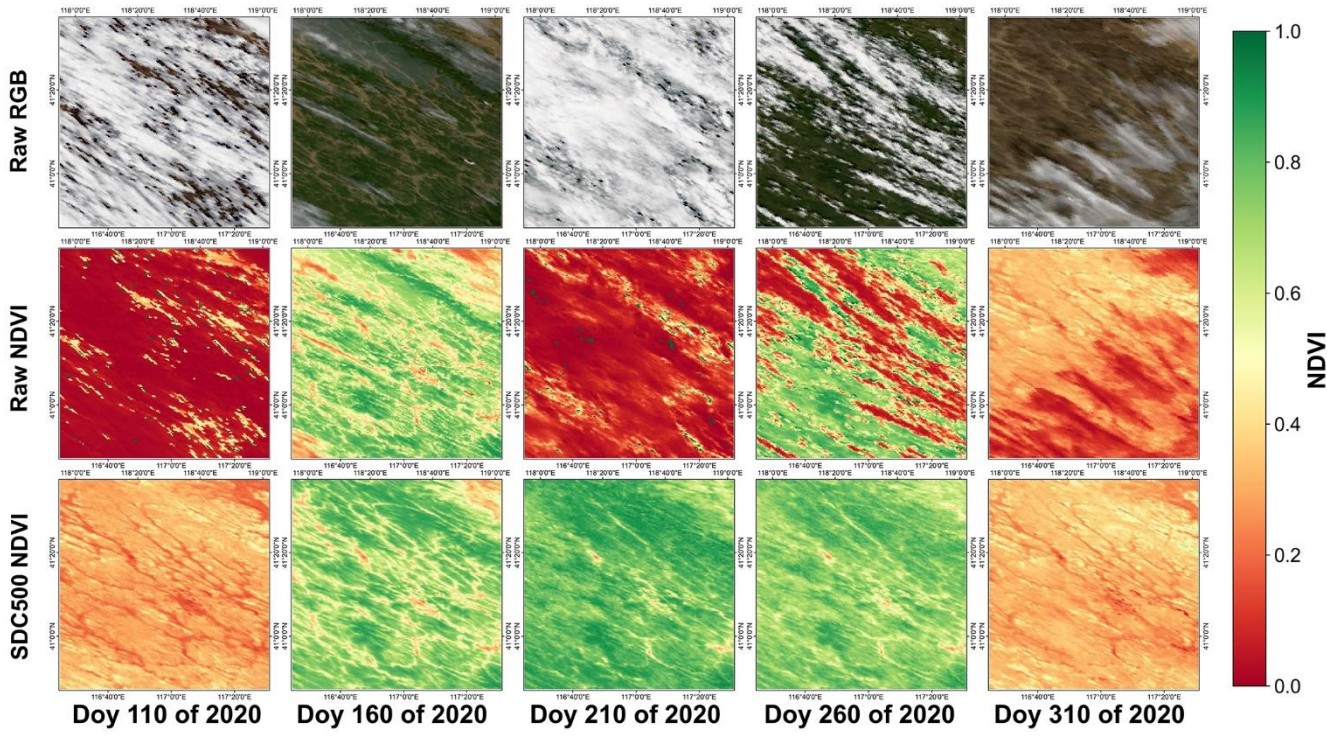


**Figure 12: Assessment of spatial pattern on a cropland area (site S5). True-colour composite (R: Band 1; G: Band 4; B: Band 3) and Raw MODIS NDVI and SDC500 NDVI imageries on 5 dates in 2020.**





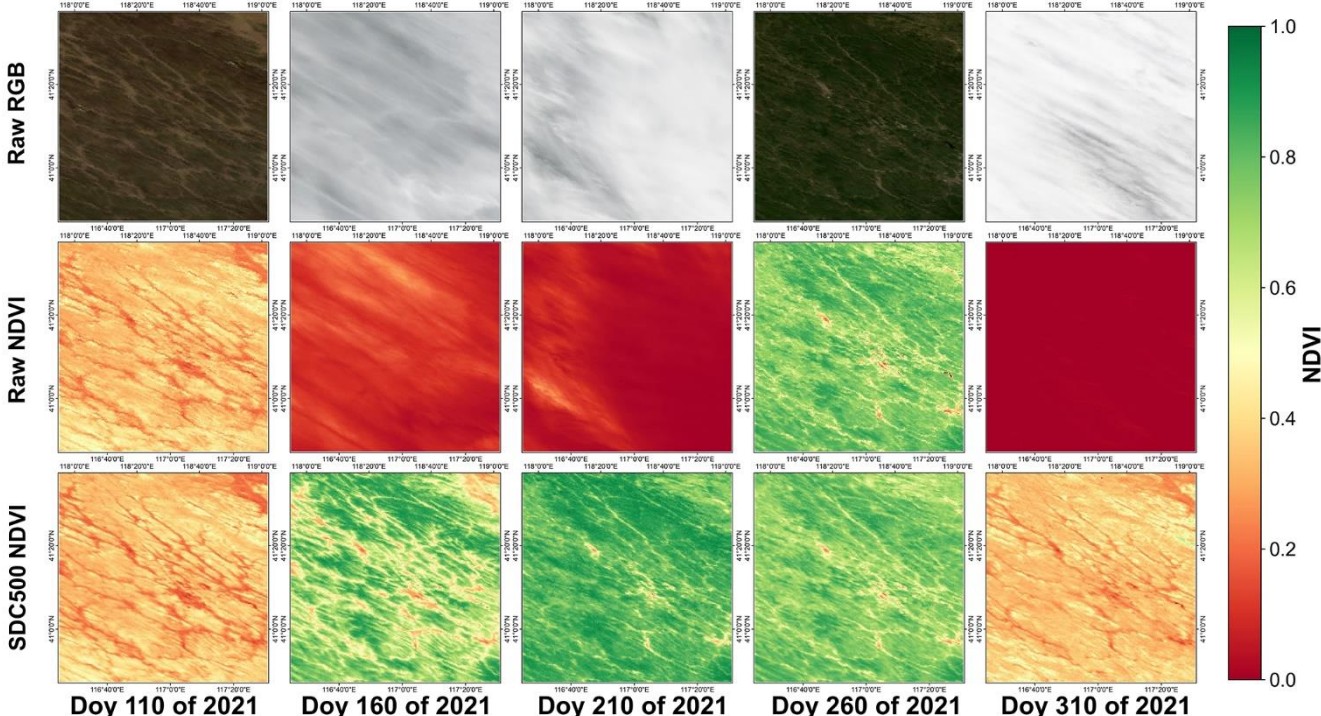

**Figure 13: Assessment of spatial pattern on a cropland area (site S5). True-colour composite (R: Band 1; G: Band 4; B: Band 3) and Raw MODIS NDVI and SDC500 NDVI imageries on 5 dates in 2021.**

Fig. 14 presents a demonstration of the raw and reconstructed reflectance and NDVI time series in S6 which is in mid-latitude evergreen broadleaf forest area. Like other site, the SDC500 dataset in S6 successfully restored the image series of evergreen broadleaf forest: the annual variation of NDVI in within range 0.3~0.5; the influence of outlier, such as DOY 100, 252, 272 in Fig. 14a, has been satisfactorily eliminated.

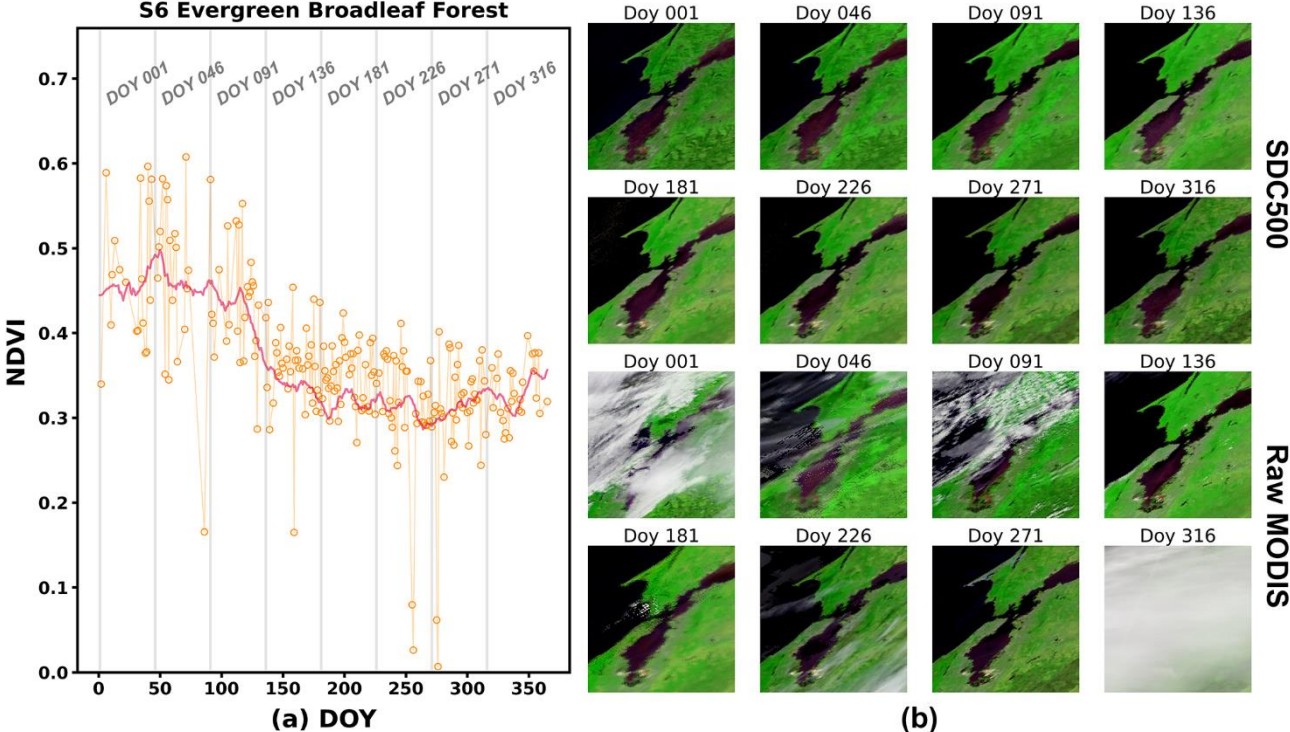

**Figure 14: Performance of SDC in site S6. (a) The NDVI curves for the central pixel, the orange points indicate the valid MODIS observation and the red line indicates the SDC500 results. (b) The spatial pattern of image blocks of 200\*200 pixels centred around**
**the site (R: Band 1; G: Band 2; B: Band 3).**

### 4.5 SDC500 in snow-dominant areas

The SDC500 demonstrates excellent capability in dealing with snow and ice cover. Fig. 15 illustrates its accurate representation of the melting and snowfall processes while retaining the intricate features of needleleaf forests hidden beneath snow and ice.
Even in images where both snow and vegetation are present, the SDC maintains spatial continuity effectively. In the Arctic region depicted in Fig. 16, the SDC500 successfully captures the short melting process in summer (DOY 180~270), showing its ability to document the variations in snow and ice cover over time.

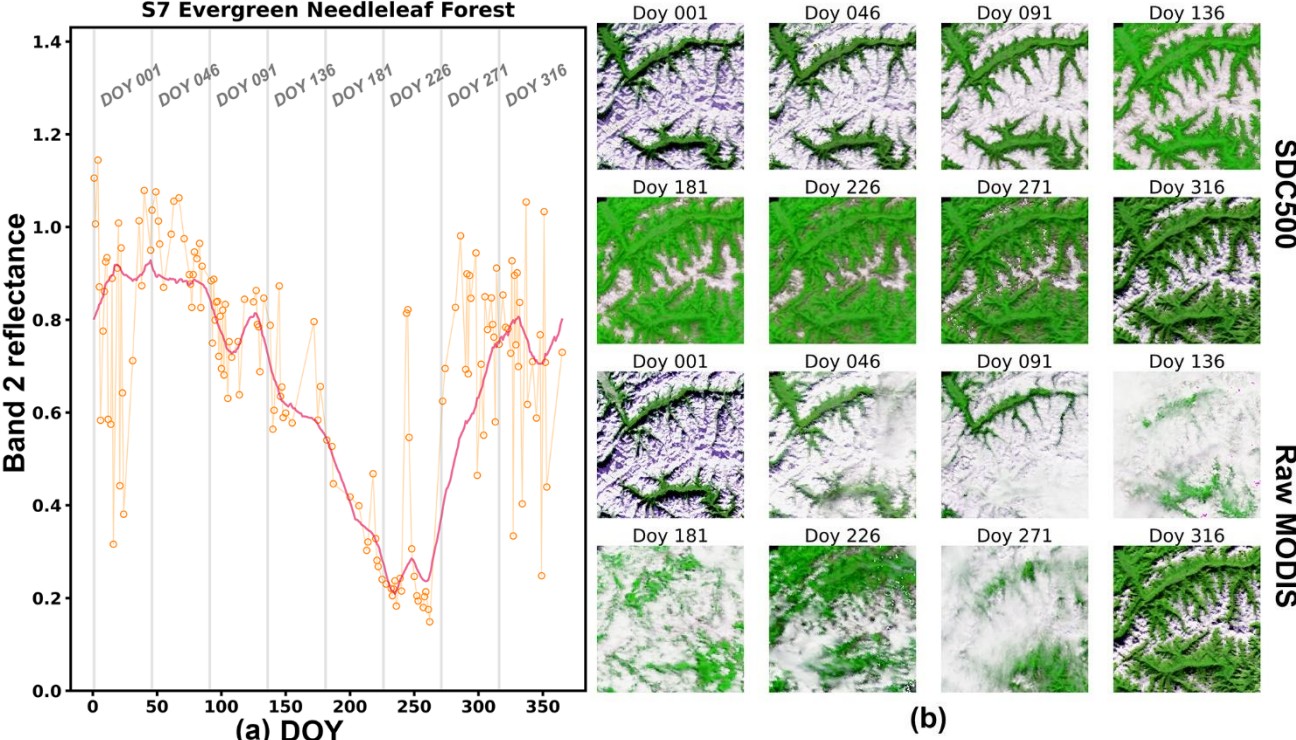

**Figure 15: Performance of SDC in site S7: (a) The band 2 reflectance curves for the central pixel, the orange points indicate the valid MODIS observation and the red line indicates the SDC500 results. (b) The spatial pattern of image blocks of 200*200 pixels centred around the site (R: Band 1; G: Band 2; B: Band 3).**



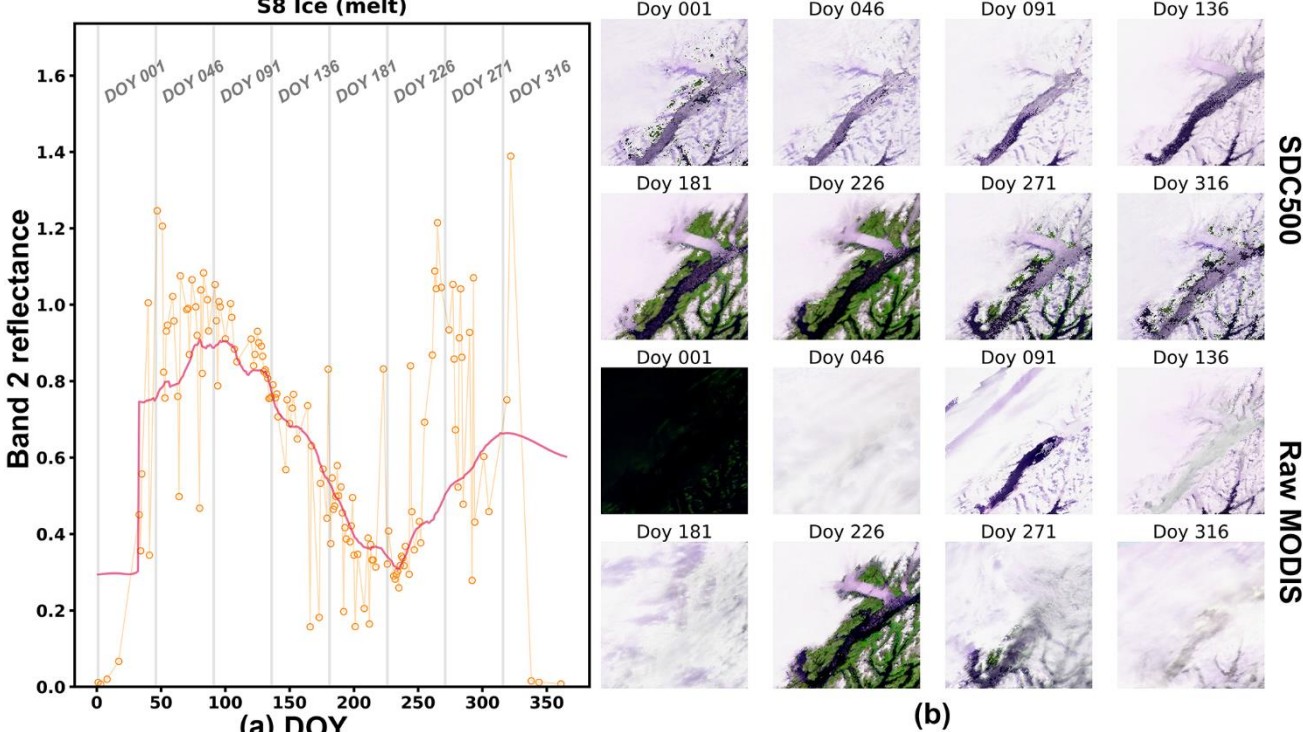

**Figure 16: Performance of SDC in site S8: (a) The band 2 reflectance curves for the central pixel, the orange points indicate the valid MODIS observation and the red line indicates the SDC500 results. (b) The spatial pattern of image blocks of 200*200 pixels centred around the site (R: Band 1; G: Band 2; B: Band 3).**



## 5 Validation and Discussion

### 5.1 Quantitative assessment of the BRDF correction method

In this study, we perform BRDF correction using land cover information from the MCD12Q1 dataset and kernel model parameters from the MCD43A1 dataset. Thus, the uncertainty in these two datasets may result in uncertainty in the BRDF corrected result. As there is no established ground-truth dataset for land surface BRDF at the 500m resolution, we evaluate the uncertainty of our proposed BRDF correction method through two approaches: 1) a visual comparison of the MODIS observations before and after the BRDF correction, and 2) a quantitative comparison with the MODIS nadir BRDF-adjusted

reflectance dataset MCD43A4.

Fig. 17 presents a comparison of the time series of raw MODIS observations and BRDF correction results in 4 typical sites. In Fig. 17, the raw clear observations are directly extracted from the MOD09GA dataset, corresponding to the various sun/view geometry of the actual satellite overpass, the BRDF-corrected results are standardized to nadir view angle and sun angle of 10:30 local time. As shown in Fig. 17, the BRDF-corrected results become more consistent with each other after

normalization, with reduced noise from biased marginal values and to some extent alleviated abnormal fluctuations. We also noticed that there are still remanent fluctuations in the normalized reflectance series, this is partly due to the reduced spatial resolution of BRDF parameters in the a priori database, and partly due to other sources of uncertainties such as geometric alignment error and atmosphere correction error.

In addition, we conducted quantitative validation between the proposed BRDF-corrected results and the MODIS nadir

BRDF-adjusted reflectance dataset MCD43A4 in the four tiles (h11v09, h20v10, h27v05, and h16v02) corresponding to the typical sites S9, S10, S11, and S12. For each MODIS tile, a regular grid of 12*12 pixels was selected as sample pixels. The raw MODIS observations in 2010, along with their corresponding normalized values, were validated against the MODIS NBAR product MCD43A4. The results, shown in Fig. 18, demonstrate that the raw MODIS reflectance has a remarkable difference with the MCD43A4 product, with an RMSE of 0.275 and a bias of 0.1721; in comparison, our BRDF correction

method is in good agreement with the MCD43A4 product, with an RMSE of 0.056 and a bias of -0.0085. As SDC500 and MCD43A4 are derived using different methods but are consistent with each other, the consistency indicates that they both provide accurate results.

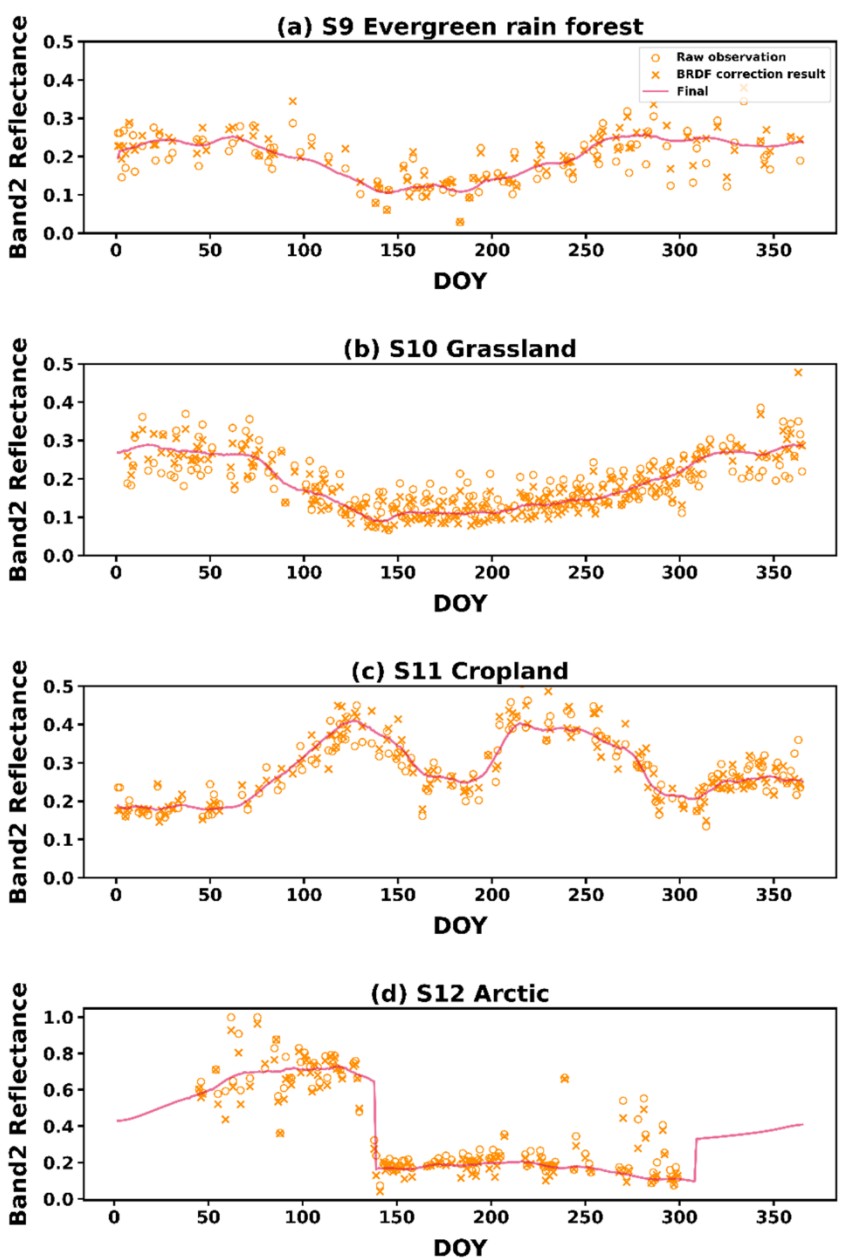

**Figure 17: Example of series of raw MODIS observations, BRDF corrected observations, and the final smoothed series in four typical sites: Site S9 Evergreen rain forest (a), Site S10 Grassland (b), Site S11 Cropland (c), and Site S12 Arctic (d).**

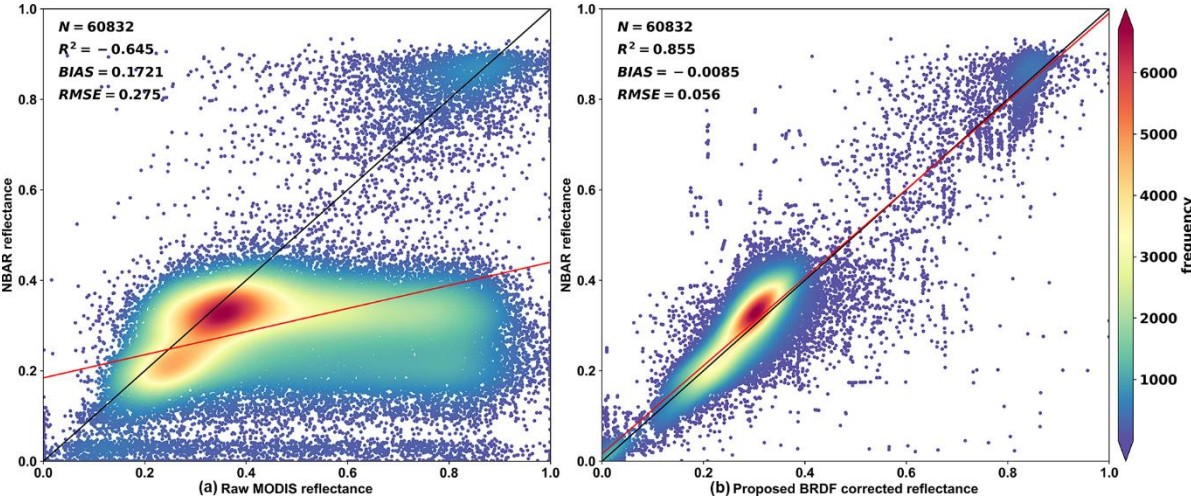

**Figure 18: Quantitative evaluation of our proposed BRDF correction method: (a) raw MODIS reflectance vs. MODIS NBAR**
**product, and (b) SDC500 dataset vs. MODIS NBAR product.**

**5.2 Quantitative assessment of the gap-filling method**

An essential module in the data processing algorithm is outlier detection and gap filling. In extreme situations of long-lasting cloud coverage, which commonly happens in tropical and monsoon areas, the gap-filling strategy basically determines the shape of the final output time series. In the past, it is common practice in time series reconstruction that the invalid values

should be masked out and filled with linear interpolation of valid values before applying the smoothing filter (e.g., the SG filter) to the time series. In this study, we use a combination of phenology-based and spline-based fitting to fill the gaps. Besides, during the time series fitting, outlier values are detected and masked. As the outlier values are most likely incorrectly flagged cloud or cloud shadow pixels, the exclusion of them can further promote the stability of the reconstruction result.

To demonstrate the performance of our proposed gap-filling algorithm in comparison to the linear gap filling. We select

an extreme case in the monsoon season in South Asia. Fig. 19a shows an area of 600*1000 pixels in MODIS tile h25v07, which locates in central India. The monsoon brings heavy cloud and precipitation in June and July every year, and leave almost no clear observation for optical remote sensing satellites in these two continuous months. Fig. 19b presents the average cloud coverage of this area in each Julian day from 2000 to 2022, as well as the cloud coverage in the year 2019. We can see that there is a rare case of a clear window during Julian day 196 to 198 in 2019, which can serve as a reference truth value to

validate the output of the time series reconstruction algorithm. From Fig. 19a it is found that the clear area is only in the centre part of the image, the surroundings are still contaminated with clouds. So, we manually outlined the clear area as a region of interest (ROI) to derive validation statistics.



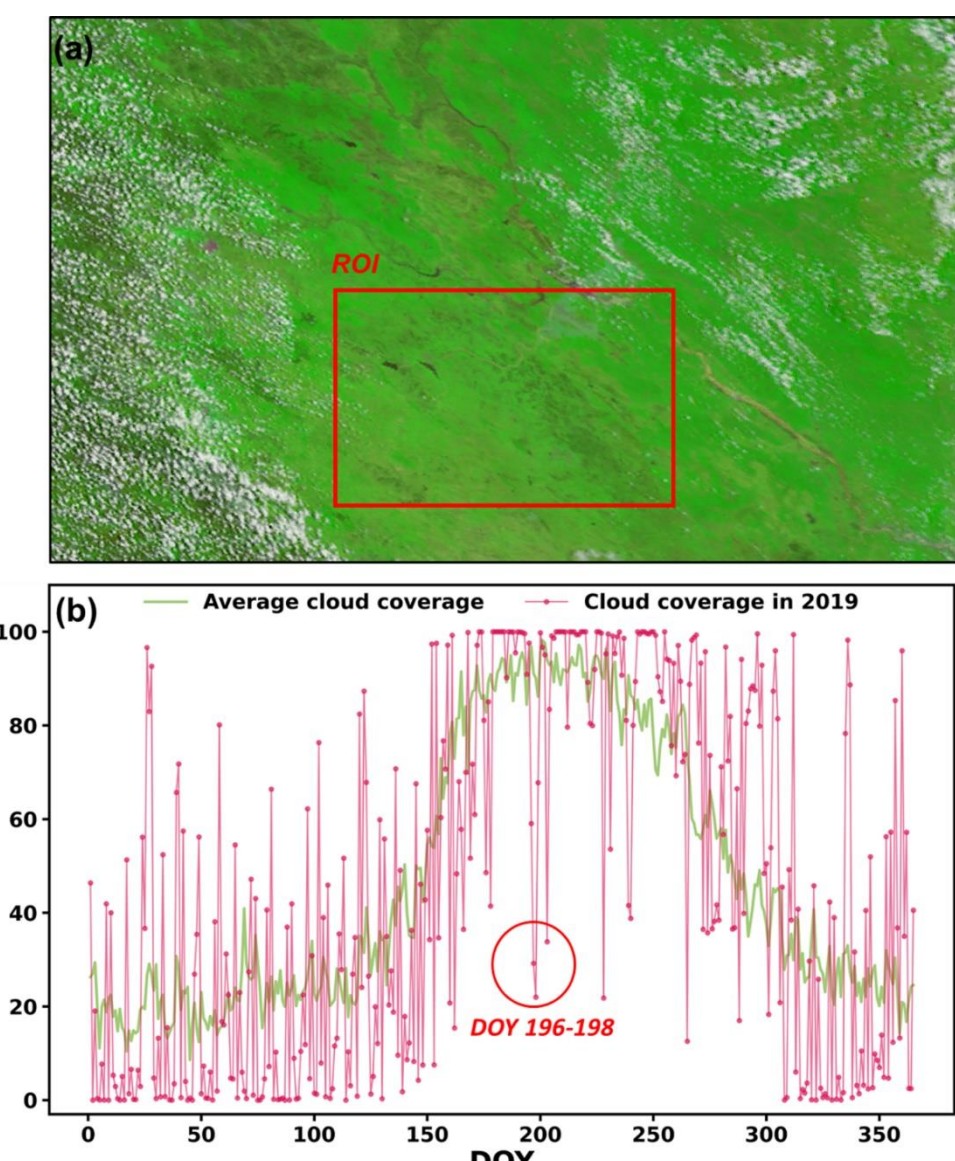

**Figure 19: Study area of quantitative validation. (a) Original MODIS observation on 2019/197, in tile h25v07, line 1-600, column 801-1800. (R: Band 1; G: Band 2; B: Band 3) The region of interest (ROI) outlined by the red box indicates the clear area to derive validation statistics. (b) Average cloud coverage in the period of 2000-2022 and cloud coverage in 2019.**




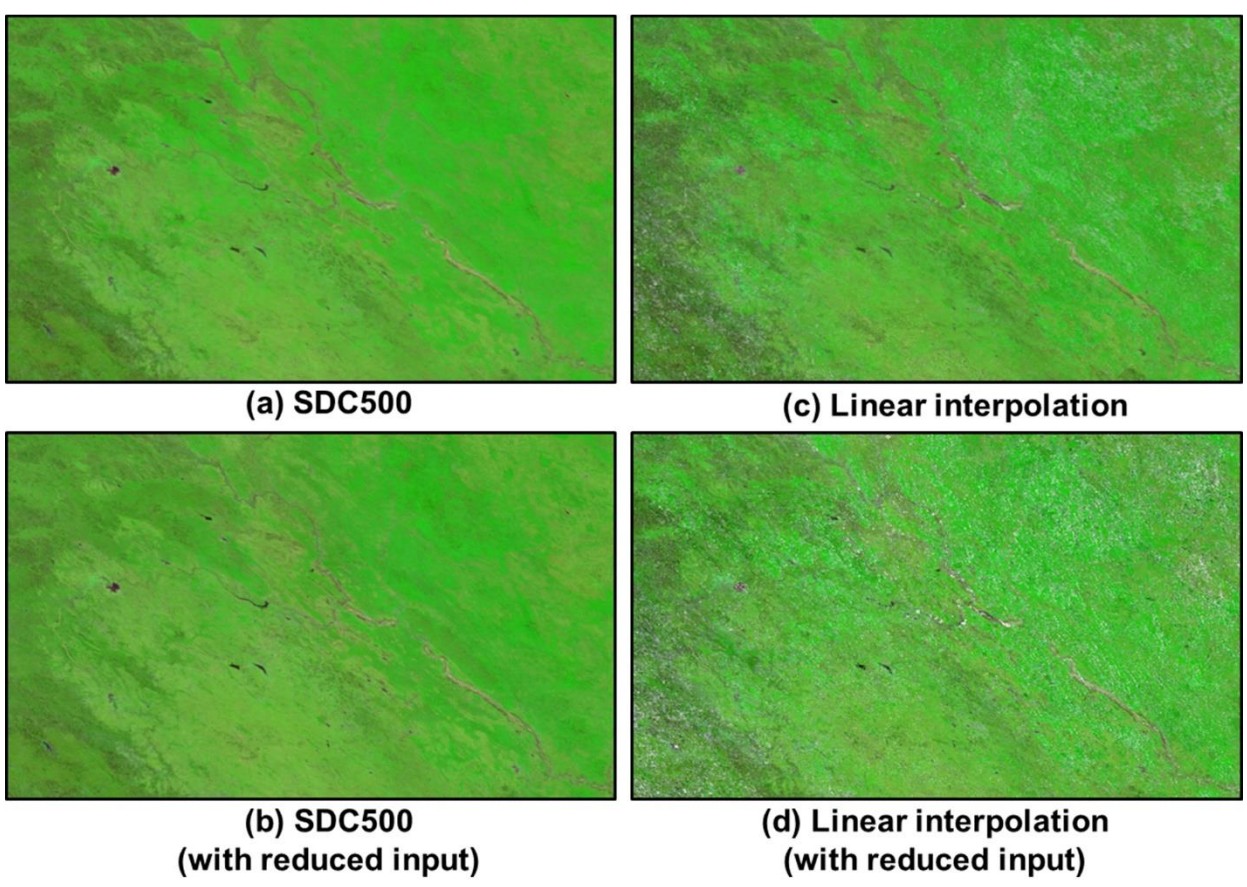

**Figure 20: Reconstructed results on 2019/197 in four cases: (a) SDC500 with all valid input data (case 1); (b) Linear interpolation with all valid input data (case2); (c) SDC500 with reduced input data (case 3); (d) Linear interpolation with reduced input data (case 4)**

**Table 2: Quantitative validation of Linear interpolation and SDC500 in different bands.**

| Band | Linear Interpolation | | | SDC500 | | |
|---|---|---|---|---|---|---|
| | *RMSE* | *MAE* | *CC* | *RMSE* | *MAE* | *CC* |
| Band1 | 0.0698 | 0.0476 | 0.1698 | 0.0502 | 0.0422 | 0.4514 |
| Band2 | 0.0867 | 0.0718 | 0.5465 | 0.0483 | 0.0382 | 0.6261 |
| Band3 | 0.0725 | 0.0477 | 0.0092 | 0.0395 | 0.0256 | 0.2043 |
| Band4 | 0.0680 | 0.0464 | 0.0443 | 0.0397 | 0.0281 | 0.3091 |
| Band5 | 0.0978 | 0.0816 | 0.4553 | 0.0615 | 0.0532 | 0.7113 |
| Band6 | 0.1407 | 0.1165 | 0.1266 | 0.0760 | 0.0705 | 0.7121 |
| Band7 | 0.0676 | 0.0537 | 0.4705 | 0.0719 | 0.0664 | 0.7057 |
| Total | 0.0844 | 0.0631 | 0.8174 | 0.0496 | 0.0430 | 0.9574 |



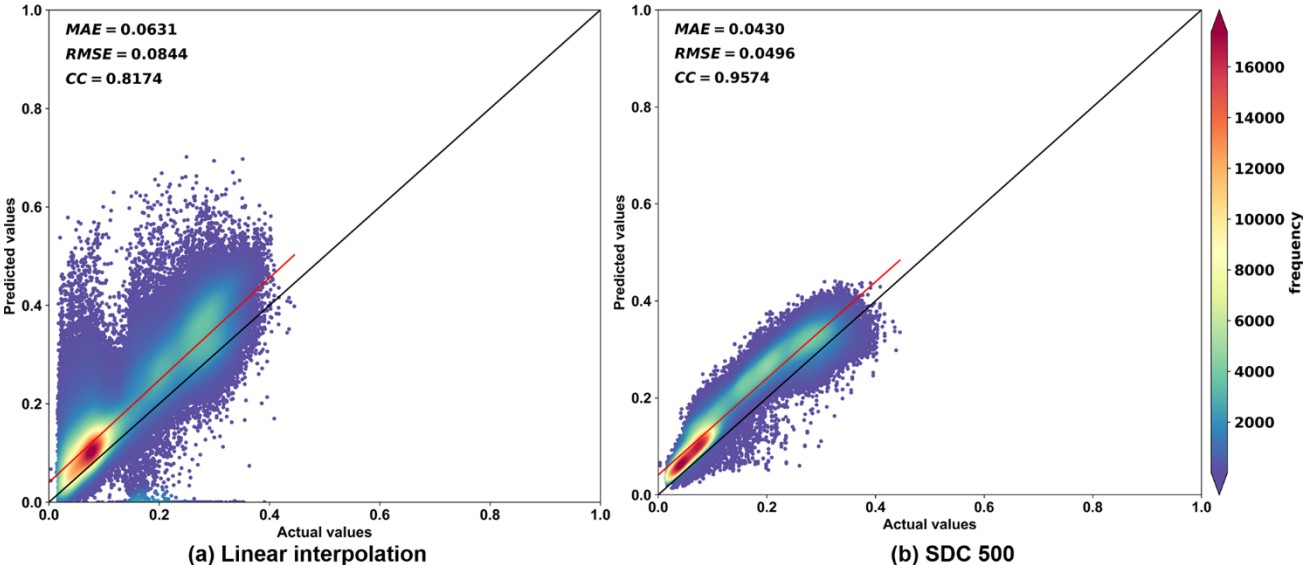

**Figure 21: Quantitative comparison between actual values and predicted values of Linear interpolation (a) and SDC500 (b).**

We implemented the algorithm in 4 cases to reconstruct the image series in 2019. Case 1 represents the operational run of the SDC500 gap-filling and smoothing algorithm, with all the available MODIS observations as input; case 2 is the same algorithm with an imaginary scenario of reduced input in which MODIS observations in Julian days 195 to 199 are excluded

from the input dataset, the purpose of this scenario is to simulate a long gap which occurs in other years; case 3 is a comparison case in which linear gap filling + SG filter smoothing is applied to the same input data as in case 1; case 4 is a comparison case in which linear gap filling + SG filter smoothing is applied to the reduced input data as in case 2. Fig. 20 illustrates the reconstructed images of the 4 cases, the good result of Fig. 20a and Fig. 20b indicate that the SDC500's gap-filling results exhibit improved spatial continuity and greater robustness with reduced input. In terms of performance evaluation metrics,

lower values of root-mean-square error (RMSE) and mean absolute error (MAE), as well as a higher correlation coefficient (CC), indicate better results. Table 2 and Fig. 21 provide the quantitative comparison between SDC500 and the result of linear interpolation. The SDC500 achieves a RMSE of 0.0496, a MAE of 0.0430, and a CC of 0.9574, demonstrating superior accuracy compared to the result of linear interpolation, which yields an RMSE of 0.0844, an MAE of 0.0631, and a CC of 0.8174.

**5.3 Performance of SDC500 in capturing rapid disturbance**

The SDC500 product provides high-resolution observations with accurate characterization of the long-term phenology of ecosystems, while also retaining the useful localized feature of rapid disturbances. In this study, we chose to demonstrate the product's capability to detect such disturbances in two wildfire events. The first event is the largest recorded tundra fire that occurred near Anaktuvuk River (Jones et al., 2009), which was started by a lightning strike in July 2007, and spread a large

area of 1,039 square kilometres till September. The other event is a serious forest fire that occurred in Australia in July 2019, which was caused by drought and heat weave weather and raged for months. To enhance the extent of the burned area, we computed the Normalized Burnt Ratio (NBR) index, with a lower NBR indicating a higher fraction of burned area. Spatially, the SDC500 was able to capture the continuous spreading pattern of the wildfires, as demonstrated in Fig. 22 and Fig. 23. Temporally, the NBR decreased sharply during the fires and returned to normal values after the Australia forest wildfire, as

seen in Fig. 23b, while remaining low after the Arctic tundra wildfire, as shown in Fig. 22b.

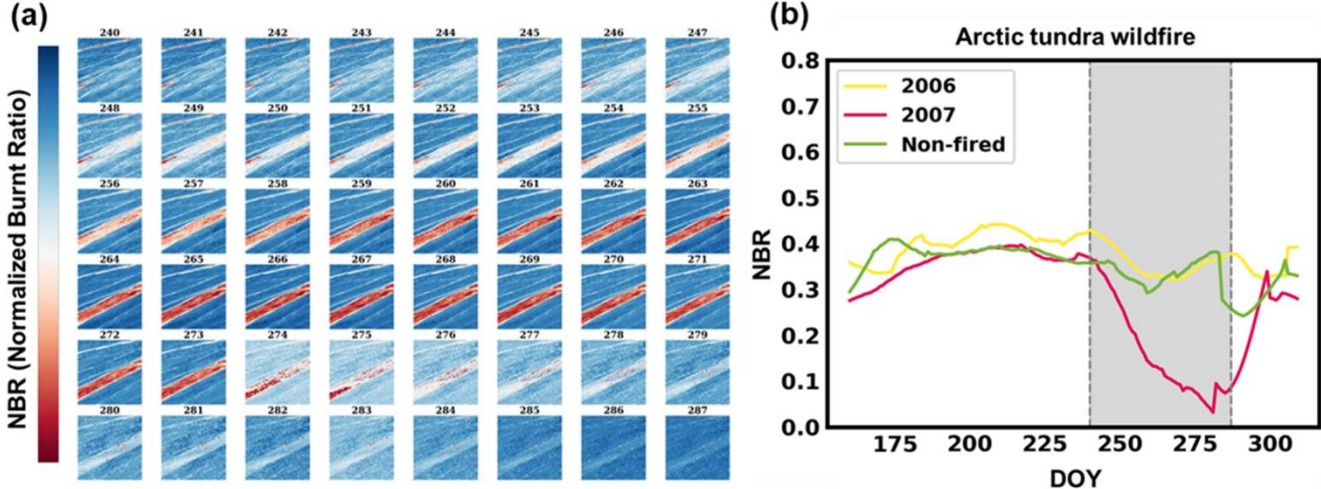

**Figure 22: The performance of SDC500 during the Arctic tundra wildfire in 2007 summer. (a) The spatial pattern of NBR. (b) The NBR series for a central pixel of burned area in 2006(before burn) and 2007(burn), and a non-fired pixel in 2007.**



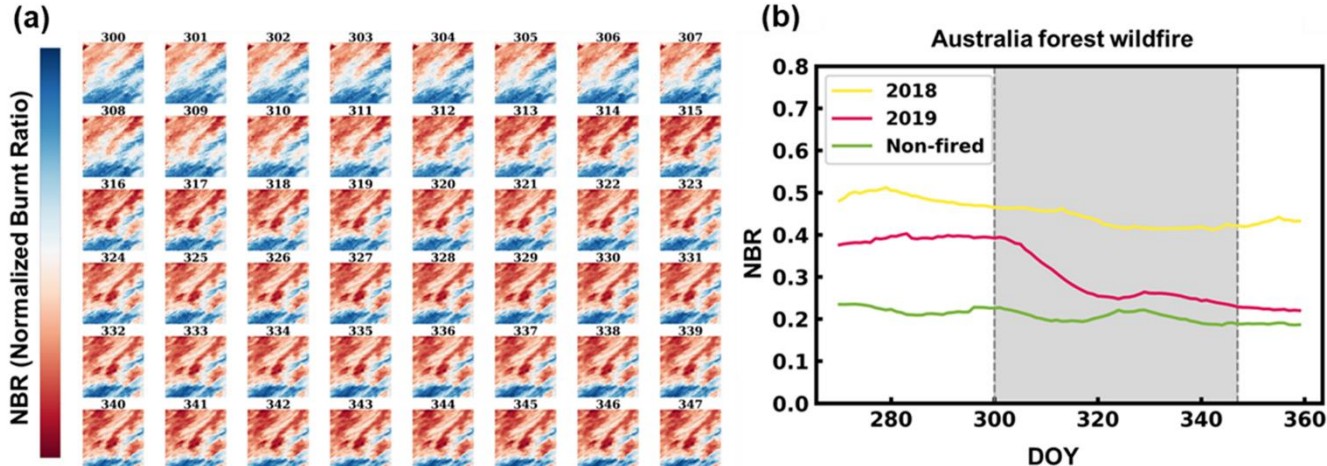

**Figure 23: The performance of SDC500 during the Australia forest wildfire. (a) The spatial pattern of NBR. (b) A central pixel of burned area in 2018(before burn) and 2019(burn), and a non-fired pixel in 2019.**

## 5.4 Limitations of the proposed SDC500 product

This study proposes an advanced framework for reconstructing the spatiotemporally seamless MODIS surface reflectance
dataset. The preliminary evaluations demonstrate the effectiveness of the SDC500 in addressing the challenge of missing key points, filling isolated and continuous gaps, as well as gaps at peaks and valleys. Moreover, the SDC500 successfully reduces noise from biased marginal values and eliminates abnormal fluctuations caused by observation conditions rather than true land surface dynamics.

Despite the satisfactory results achieved by the SDC500 dataset generated using the proposed methodology, several issues
still need to be addressed and improved in future research. Firstly, the BRDF correction relies heavily on a priori parameters derived from the MCD43A1 dataset and the empirical RTLSR model. However, the RTLSR model, which was originally designed for vegetated land pixels, may not perform well in snow/ice-covered surfaces and water bodies, leading to higher uncertainty in the SDC500 for these areas compared to vegetated and arid regions. The current granularity used to derive the statistics of BRDF parameters is 100 km × 100 km, which may be too coarse and result in some loss of accuracy. However,
this coarse granularity was chosen to ensure robust statistics. Future improvements should focus on exploring enhancements to the BRDF model and parameterization.

Secondly, in terms of prolonged gaps caused by cloudy days, there are still extreme cases that cannot be adequately filled using the proposed algorithm. Visual inspection has identified these extreme cases in MODIS tiles h09v07, h10v08, h10v09, h28v08, h28v09, and h29v09, mostly in mountainous terrain within tropical rainforest climate zones. In these areas,
the topographic cloud formed by the terrain effect persists throughout the year, resulting in almost no clear satellite images.



Specialized treatments are required for these pixels, which are not included in the current SDC500 algorithm. As compensation, the quality assessment (QA) flags of these pixels are set as low quality to alert data users to exercise caution when utilizing this data.

Another noteworthy case is the transition between snow-covered and snow-free states in high-latitude area. As described in section 3.5, the observations in a pixel's time series are divided into two groups: snow-covered and snow-free, and processed separately. When these two results are combined, the final series may exhibit discontinuities around the switching date. Furthermore, the current algorithm struggles to robustly reconstruct images near the date of surface status change, presenting a challenging research topic.

In general, the process of image reconstruction involves altering the original values of sensor observations and replacing them with approximated values. Therefore, the reconstructed dataset is primarily intended to support large-scale statistical research or discrete applications such as land cover classification, rather than small-scale or in-depth research such as model tuning and validation. In the past, people tend to be overcautious to share the reconstructed dataset, fearing to mislead data users. However, the demand for seamless data cubes has prompted us to share the SDC500 dataset and provide an opportunity for evaluation and improvement by the wider public.




## 6 Conclusions

The state-of-art Moderate Resolution Imaging Spectroradiometer (MODIS) surface reflectance products suffer from temporal and spatial gaps, which make it difficult to characterize the continuous variation of the terrestrial surface. There are two challenges in reconstructing spatiotemporal seamless surface reflectance. First, the intrinsic inconsistency of observations

owing to various sun/view geometry. Second, the prolonged missing values result from heavy cloud coverage, especially in monsoon season or polar night.

In this study, we have established a framework to address these two challenges. And the first global 500 m daily Seamless Data Cubes (SDC500) of surface reflectance was produced based on MODIS products, covering the period from 2000 to 2022. The proposed framework contains the generation of a land cover-based *a priori* database, BRDF correction, outlier detection,

gap filling, and smoothing. In consideration that the change of surface reflectance is abrupt in the snow/thaw process, the time sequence is divided into the snow-covered and snow-free parts and processed separately.

The quantitative assessment showed that proposed gap-filling results have a RMSE of 0.0496 and a MAE of 0.0430. In addition, the proposed BRDF correction results showed a RMSE of 0.056 and a bias of -0.0085 when compared with MODIS NBAR products, indicating the acceptable accuracy of both products. Furthermore, assessment of SDC500 at 12 sites

worldwide with different land cover demonstrated its robust performance in tropical and subtropical areas (Sites S1, S2), acrid areas (Sites S3, S4), temperate climate areas (Sites S5, S6), and snow-dominant areas (Sites S7, S8). From a temporal perspective, the SDC500 eliminates abnormal fluctuations while retaining the useful localised feature of rapid disturbances. From a spatial perspective, the SDC500 shows satisfactory spatial continuity.

The SDC500 product can serve as fundamental input for ecological, agricultural, environmental applications and

quantitative remote sensing studies, eliminating the time-consuming and labour-intensive preprocesses that is typically required. In addition, the SDC500 dataset will be a necessary step towards the generation of a new version of 30 m resolution seamless data cube (Liu et al., 2021), which will greatly improve the global land use/land cover classification accuracy as well as capture its dynamics.

## Code/Data availability

The SDC500 is available at: http://data.starcloud.pcl.ac.cn/resource/27 or https://doi.org/10.12436/SDC500.27.20230701 (Liang et al., 2023). All users are welcome to freely download it. This dataset covers all global land surface, as well as part of shallow sea areas, in the temporal range of 2000 - 2022. The spatial resolution is 500m, and temporal resolution is 1 day. The dataset is stored in GEOTIFF format and Sinusoidal projection. The dataset is composed of 8 bands. The first 7 bands correspond to surface reflectance in the MODIS band 1 to band 7, with a scale value of 0.0001 to convert DN value to

reflectance. The 8th band is the quality assessment (QA) flag, in which the overall quality is indicated in the lowest 2 bits, i.e., bits 0~1, with 00 represents the best credibility and 11 represents the worst credibility; and bit 6 is an indicator of snow status,

with value 0 indicate snow-free and value 1 indicate snow; bit 7 is an indicator of polar night, with value 0 means normal and value 1 indicate the pixel is in polar night (solar zenith angle larger than 82 DEG at 10:30 local time) in the specified date.

The GEE code used to derive BRDF parameters from MCD43A1 and MCD12Q1 products can be accessed at:
https://code.earthengine.google.com/363b4d94090048f9e28103ad3efebfdf.

**Author contributions.** LQ conceptualized and supervised the project. LQ, LX, WJ designed the workflow of methodology to product the dataset. LQ, LX, WJ, and CS conducted the work in data acquisition and processing. LQ, LX performed data analysis and prepared the manuscript. WJ, CS and GP reviewed and edited the draft. All authors contributed to the
interpretation of the results.

**Competing interests.** The authors declare that they have no known competing financial interests or personal relationships that could have influenced the work reported in this study.

**Acknowledgments.** This research was partially supported by two NSFC grants (42090015; 42071400) and the Major Key Project of PCL. We would like to thank NASA for providing MODIS data products.

**Disclaimer.** Publisher's note: Copernicus Publications remains neutral with regard to jurisdictional claims in published maps and institutional affiliations.




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
