# Peer review of "Global 500 m seamless dataset (2000-2022) of land surface reflectance generated from MODIS products"

_Earth System Science Data, 2023_

## Author Comment (AC1)

First of all, we want to thank all the reviewers for their careful reviews. The comments and suggestions have helped us to improve the paper a lot. We have made the revisions carefully according to the review comments and hope the reviewers find them satisfactory. Below are our responses to the comments.

(p.x, lines xx-xx) denotes the page and lines in the revised manuscript (Track Changes – All Markup).

**Reviewer #1**

**Comment 1: This study attempts to create a gap-filled (seamless) surface reflectance datasets from the corresponding MODIS products. The work is important, the methodology is largely OK (but with caveats indicated below), and the writing of the paper is clear. Overall I think the paper and the data products have the potential to be published. My main concern with the current paper is that it lacks a quantitative description of the uncertainties associated with the gap-filling method and thus the final results. At least these uncertainties should be discussed in the paper.**

Response:

Thank you for the constructive comments which have greatly helped us in revising and improving the manuscript. We will respond your comments item by item. We admit that the quantitative analysis of the uncertainties associated with the gap-filling method is weak in this manuscript. Quantitative uncertainty analysis is very difficult in this research because of the complexity of data processing steps, and the lack of quantitative uncertainty information about the input data. We hope the anonymous reviewer can understand the limitations in our current research work.

We have rewrite section 5.4, adding an analysis about the QA (quality assessment), as well as more discussions on the possible uncertainties in the SDC500 dataset. However, our QA is more like a relative indicator, rather than a quantitative one. (p.34, lines 520-563)

Although there lacks suitable surface reflectance to directly validate SDC500, it is still possible indirectly evaluate the dataset with applicational results derived from SDC500. In one of the example applications, we've conducted a quantitative evaluation of the phenology parameters extracted from SDC500 against the phenology parameters extracted from MCD13C1, which is a NDVI product in 16-day temporal resolution, generated from MODIS observations by NASA. The ground truth (reference) data for phenology is calculated from carbon flux measurement of 48 sites in the Northern Hemisphere from Fluxnet2015 dataset.

To compare the phenology extracted from SDC500, MCD13C1 and reference data. We calculated the Start of Season (SOS) from MOD13C1, SDC500 and reference ground truth data for evergreen forest (EF), deciduous forests (DF)), mix forest (MF,) and shrub-grassland (SH-GRA). Among all the four situations, SOS extracted from SDC-NDVI data showed more similarity with the reference data than the results from the MOD13C1 data, with higher R2 and lower RMSD (Figure S1).

[Figure]

**Figure S1** Comparison of phenology extracted from SDC500, MCD13C1 and reference data for four different land cover types.

The details of this experiment are as follows:

**1.1 Distribution of flux tower observation sites.**

The FLUXNET2015 (https://fluxnet.fluxdata.org/data/fluxnet2015-dataset/) dataset provides carbon fluxes at multiple time dimensions (half-hourly, hourly, daily, weekly, monthly, and yearly). In this study, we used the daily Gross Primary Productivity (GPP) products (GPP _ DT _ VUT_ REF) from FLUXNET2015 dataset to extract phenology. We selected a total of 48 flux sites in the Northern Hemisphere (>40°N) with quality control ("NEE_VUT_REF_DAY_QC" > 0.70) from the FLUXNET2015 dataset, representing evergreen forest (38 records), deciduous forest (71 records), mixed forest (26 records), and shrubs and grasslands (42 records). Due to significant human impact on croplands, we did not include cropland sites in the ground-based validation phenology sites.

[Figure]

**Figure S1.1.** Distribution of flux tower observation sites. The inset image in the bottom left corner shows the proportion of each vegetation type. EF (Evergreen forest), DF (deciduous forests), MF (mix forest), SH (Shrubland), GRA (grassland), and CRO (cropland) respectively. The time span indicates the duration of continuous records collected at each flux observation site.

**1.2 Calculation of SOS and EOS**

Firstly, we calculated NDVI from SDC and MOD13C1. For the FLUXNET2015 data, GPP sequence was directly used to represent the NDVI sequence.

To eliminate the influence of bare soil, gravel, and sparse vegetation, we first excluded pixels with an annual average NDVI < 0.1 (Piao et al. 2006). Given that the SDC-NDVI dataset has already undergone smoothing, we solely applied the smoothing algorithm to the MOD13C1 dataset. For the MOD13C1 dataset, we used a double logistic function to fit and reconstruct the daily time series NDVI (Beck et al. 2006). This function is widely used in phenological research(Chang et al. 2019; Guo et al. 2021; Zeng et al. 2021; Guo and Hu 2022)   and is considered to be the optimal surface phenological retrieval method (Eq. (1)):

$$NDVI(t) = \alpha_1 + (\alpha_2 - \alpha_1) \times \left( \frac{1}{1+\exp(-\alpha_3 \times (t-\alpha_4))} + \frac{1}{1+exp(a_5 \times (t-a_6))} \right) \qquad (1)$$

Where $NDVI(t)$ means the fitted NDVI at day t; $\alpha_1$ and $\alpha_2$ mean minimum and maximum value of NDVI for each year; $\alpha_3$ and $\alpha_5$ mean increase and decrease curve coefficient at the inflection point; $\alpha_4$ and $\alpha_6$ mean the initial estimates of the inflection points.

Finally, the land surface phenology was retrieved using the dynamic threshold method. First, the $R(t)$ was calculated using Eq.(2):

$$R(t) = \frac{NDVI_t - NDVI_{min}}{NDVI_{max} - NDVI_{min}} \qquad (2)$$

Where $NDVI_t$ represents daily NDVI pixel value at date t which can be either from SDC-NDVI or from reconstructed MOD13C1; $NDVI_{max}$ represents the maximum NDVI for the whole year; The minimum value in the dynamic threshold method can cause significant fluctuations in phenological extraction results due to the influence of factors such as snow cover and bare soil. In this study, we fixed the minimum value at 0.1 to improve the stability of data processing and the accuracy of results, based on previous research(Yu, Luedeling, and Xu 2010). The phenological parameter SOS (EOS) is defined as the first date of the year when the   $NDVI_{ratio}$ reaches (drops below) 0.5 during the ascending (descending) period. This approach has been widely applied and the retrieval results are within an acceptable range(Dong et al. 2023; Du et al. 2022).

Reference:

Beck, Pieter S. A., Clement Atzberger, Kjell Arild Høgda, Bernt Johansen, and Andrew K. Skidmore. 2006. "Improved monitoring of vegetation dynamics at very high latitudes: A new method using MODIS NDVI." Remote Sensing of Environment 100 (3):321-334. doi:10.1016/j.rse.2005.10.021.

Chang, Qing, Xiangming Xiao, Wenzhe Jiao, Xiaocui Wu, Russell Doughty, Jie Wang, Ling Du, Zhenhua Zou, and Yuanwei Qin. 2019. "Assessing consistency of spring phenology of snow-covered forests as estimated by vegetation indices, gross primary production, and solar-induced chlorophyll fluorescence." Agricultural and Forest Meteorology 275:305-316. doi:10.1016/j.agrformet.2019.06.002.

Dong, Lingwen, Chaoyang Wu, Xiaoyue Wang, and Na Zhao. 2023. "Satellite observed delaying effects of increased winds on spring green-up dates." Remote Sensing of Environment 284:113363. doi:10.1016/j.rse.2022.113363.

Du, Hui, Mei Wang, Yuxin Liu, Mengjiao Guo, Changhui Peng, and Peng Li. 2022. "Responses of

autumn vegetation phenology to climate change and urbanization at northern middle and high latitudes." International Journal of Applied Earth Observation and Geoinformation 115:103086. doi:10.1016/j.jag.2022.103086.

Guo, Mengdi, Chaoyang Wu, Jie Peng, Linlin Lu, and Shihua Li. 2021. "Identifying contributions of climatic and atmospheric changes to autumn phenology over mid-high latitudes of Northern Hemisphere." Global and Planetary Change 197:103396. doi:10.1016/j.gloplacha.2020.103396.

Guo, Jinting, and Yuanman Hu. 2022. "Spatiotemporal variations in satellite-derived vegetation phenological parameters in Northeast China." Remote Sensing 14 (3):705. doi:10.3390/rs14030705.

Piao, Shilong, Jingyun Fang, Liming Zhou, Philippe Ciais, and Biao Zhu. 2006. "Variations in satellite-derived phenology in China's temperate vegetation." Global change biology 12 (4):672-685. doi:10.1111/j.1365-2486.2006.01123.x.

Yu, Haiying, Eike Luedeling, and Jianchu Xu. 2010. "Winter and spring warming result in delayed spring phenology on the Tibetan Plateau." Proceedings of the National Academy of Sciences 107 (51):22151-22156. doi:doi:10.1073/pnas.1012490107.

Zeng, Zhaoqi, Wenxiang Wu, Quansheng Ge, Zhaolei Li, Xiaoyue Wang, Yang Zhou, Zhengtao Zhang, Yamei Li, Han Huang, and Guangxu Liu. 2021. "Legacy effects of spring phenology on vegetation growth under preseason meteorological drought in the Northern Hemisphere." Agricultural and Forest Meteorology 310:108630. doi:10.1016/j.agrformet.2021.108630.

**Comment 2: Line 167-170 (Section 3.1, Generation of the a priori database). Predicting missing values at a location from sample values at other locations is a typical problem of spatial analysis or geostatistics. There are rich methodologies to address the problem (e.g., Kriging, Sequential Gaussian Simulation, etc.) but you appear to use a very simple home-made approach. My concern with your approach is that it lacks a way to quantify the uncertainties associated with the predicted values or, more rigorously, the random spatial field. Take the example of the grid cell in a tropical desert described here, there are simply no adjacent pixels to provide information of snow, crop, or forest. Therefore the information you fill in is purely random with large uncertainties (compared to other cases). And based on my understanding, this random information (time series) might be used in the next step (Landcover-based BRDF correction), which can be problematic! Even though this could be an extreme example, the point is that your methodology is too simple and lacks necessary mathematical/statistical rigor. There are numerous curves that can make the data fitting "looks" good (e.g., artists can do it) but, for science, we must make our choice based on solid quantitative reasoning.**

Response:

Thank you for the comment. It is an aspect which we haven't paid enough attention before. In the generation of the a priori database, there is a step to gap-fill the a priori database. We have chosen a rather simple algorithm for this process, following two principles: 1) The a priori database is only used in the first step of the overall algorithm, and the information of observations are infused into the result mainly in later steps. Although the a priori database is indispensable, its influence to the final result is very small in most of the regions and seasons. So, it is our intention to choose simple process for generating the a priori database. 2) It is import that the a priori database should be stable,

which means, the gap-fill value should not be too random to go out of the possible boundary. And the current gap-fill method satisfies our demand on simpleness and stability.

In the following part, we will present the detailed reason for our "home-made" method, hereby referred as "iterative local average filter", with an example.

Figure S2a presents the global spatial distribution of the f_iso of NIR band for snow-free grasslands from the statistics of MCD43A1 product. All the black area indicates no valid statics. We can see that although grasslands is wild spread in most main continents, there are still wide gaps in its global distribution, and the typical values of f_iso are significantly different in different regions. With this distribution of samples, it is very hard to robustly derive the global-valid geostatistics model, and it is not reasonable to extrapolate values of these samples to no-sample regions such as the oceans. However, we need a seamless a priori database to simplify the programming of the overall algorithm. In this circumstance, it is tradition to choose the simple and robust gap-filling method such as nearest neighbor interpolation or average filter. Figure S2b presents the result of nearest neighbor interpolation, and Figure S2c presents the result of our iterative local average filter. We can see that Figure S2c resembles Figure S2b in pixels which are far from the groups of samples. So, the iterative local average filter method can be looked upon as a combination of nearest neighbor interpolation and average filter, and its main merit is that the extrapolation value will not exceed the original sample values in all the circumstances.

We appreciate the anonymous reviewer's suggestion to use spatial analysis mathematical tools to fill the a priori database. We have tried Ordinary Kriging interpolation, and an example result is presented in Figure S2d. We can see the values in Figure S2d become too large or too small when they are extrapolated far away from original samples. This is what we have to avoid. We can't say that our use of Kriging is all correct, as there are many ways to improve the Kriging performance. Our argument is that Kriging is not a straight forward way to fill the a priori database because of its possibility to produce out-of-boundary values.

The reviewer also mentioned the Sequential Gaussian Simulation, which is a potential mathematical tool to fill the gaps. There are also other potential mathematical tools such as the inverse distance weighting (IDW) and trend surface fitting. We will try them in our future research.

[Figure]

(a) original values

(b) nearest neighbor interpolation

(c) iterative local average filter

(d) Ordinary kriging interpolation

**Figure S2** The global spatial distribution of the f_iso of NIR band for grassland land cover. (a) original values from the statistics of MCD43A1 product; (b) gap-fill result of nearest neighbor interpolation; (c) gap-fill result of iterative local average filter; (d) gap-fill result of ordinary kriging interpolation.

**Comment 3: For the same Section 3.1, Land cover type is not the only local factor that regulates BRDF. Topographic characteristics of the terrain (e.g., slope structure) can have significant influence on BRDF as well. It should be at least discussed.**

Response: Topography is really an important factor for the normalization of land surface reflectance, and also a very difficult challenge. We have to admit that the current SDC500 algorithm does not consider topographic effect. We have added some discussion about topography in section 5.4, and added a reference article about it. (p.36, lines 540-545) (p.37, lines 557-559)

**Comment 4: Line 175-179. I don't fully understand why you'd use all the 17 curves to fit the observed time series of surface reflectance. Why not just pick the curve of the same land cover type? Or broad categories (e.g., forests or vegetation)?**

Response:

Thank you for raising this topic. We are sorry that the former version of manuscript over-simplified this description of the algorithm. Actually, the algorithm does not use all the 17 curves to fit the observed time series of surface reflectance, but uses a smaller group of curves, which is somewhat similar to broad categories. For example, if the pixel is indicated as evergreen needleleaf forest in the MCD12Q1 product, then, the a priori curves for evergreen needleleaf forest, evergreen broadleaf forest, deciduous needleleaf forest, mixed forests, closed shrubland will be candidates to fit the observed time series of this pixel; and the one curve with minimum fit error is the final adopted curve. The criterion to select candidate is that the candidate LC type should be prone to be mis-classified or actually be composing mixed-pixel with the target LC type. Table S1 present the lookup table of candidate LC types, which is used in our programing of the algorithm.

We have modified the description of algorithm in section 3.2. (p.29, lines 179-188)

Table S1  The lookup table of candidate LC types. The numbers for land cover types are:1) Evergreen needleleaf forest; 2) Evergreen broadleaf forest; 3) Deciduous needleleaf forest; 4) Deciduous broadleaf forest; 5) Mixed forests; 6) Closed shrubland; 7) Open shrublands; 8) Woody savannas; 9) Savannas; 10) Grasslands; 11) Permanent wetlands; 12) Croplands; 13) Urban and built-up; 14) Cropland/natural vegetation mosaic; 15) Snow and ice; 16) Barren or sparsely vegetated; 17) Water

| Target LC type | Possible alternative LC types | | | |
|---|---|---|---|---|
| 1 | 2 | 5 | 6 | 3 |
| 2 | 1 | 5 | 4 | 6 |
| 3 | 4 | 5 | 1 | 6 |
| 4 | 5 | 3 | 2 | 6 |
| 5 | 4 | 3 | 2 | 1 |
| 6 | 7 | 5 | 8 | 4 |
| 7 | 6 | 8 | 5 | 9 |
| 8 | 9 | 7 | 5 | 10 |

| 9 | 8 | 10 | 7 | 6 |
|----|----|----|----|----|
| 10 | 9 | 8 | 7 | 16 |
| 11 | 14 | 17 | 6 | 8 |
| 12 | 14 | 11 | 13 | 16 |
| 13 | 16 | 12 | 14 | 10 |
| 14 | 12 | 10 | 5 | 9 |
| 15 | 16 | 11 | 10 | 13 |
| 16 | 13 | 7 | 9 | 10 |
| 17 | 11 | | | |

**Comment 5: Line 55-57. "Urgent" may not be the best word to use here. After all, the MODIS instruments are > 20 years old already.**

Response: Thank you for the comment. We have changed the phasing as: "Therefore, it is desirable to generate a pre-processed seamless (gap-filled) global land surface reflectance dataset to serve as the primary input for the ecological, agricultural, and environmental applications, as well as quantitative remote sensing studies." (p.2, lines 54)

**Comment 6: Line 146. "BRDR" should be "BRDF".**

Response: Thank you for pointing out the error. We have corrected it in the revised manuscript. (p.7, lines 149)

**Comment 7: Line 157-158. The last sentence "They are roughly independent … " is confusing and can be removed.**

Response: Thank you for the comment. We have removed this sentence. (p.8, lines 160-161)

**Comment 8: Line 185.   The Greek letter "phi" on the left-hand-side of the equation should be "phi 0".**

Response: Thank you for pointing out the error. We have corrected it in the revised manuscript. (p.9, lines 194)

**Comment 9: Line 305 and Fig. 5. The gray thin lines (spline-filled) show spikes at some of the raw observations. How is it possible, for the spline base functions are smooth low-frequency functions?**

Response: Thank you for pointing out this issue. We are not clear in our former description of Fig.5. The gray thin line should be the "spline-filled values + raw values". In contrast, the light green thick line present only the "phenology filled values". In order to make the figure simple and discernible, the raw values are excluded from light green thick line. We are sorry for the confusion, and modification is made in Fig.5 in the revised manuscript. (p.16, lines 314)

**Comment 10: Figs. 5-16. What is the relationship between the SDC500 curves and the original MODIS observations? It doesn't seem to be the time-smoothed mean at locations. Again, is there a way to quantify the uncertainties associated with the smoothed curves versus the raw observations?**

Response:

Thank you for the comment. The SDC500 curves should theoretically be close to the smoothed mean of MODIS observations. However, the deviation is reasonable for the following reasons: 1) the raw MODIS observations contain BRDF effect while the SDC500 data are BRDF normalized, and this explained some of the deviation such as in Julian day 250~365 in Fig. 9a. 2) the SDC500 are reflectance data, while several of the figures in 5-16 are NDVI curves, the difference between NDVI and reflectance make these NDVI figures deviated. 3) there is an outlier detection process in the SDC500 algorithm. After the removal of outliers in raw MODIS data, the resulting SDC500 curve can deviate from the smoothed mean. The effect of outlier removal is more discernable in the snow contaminated cases of Fig. 15 and Fig. 16, in which we have to admit that out outlier detection was not very perfect during the snow/snow-free switching period.

The suggestion of "quantify the uncertainties associated with the smoothed curves versus the raw observations" is share by the other reviewers. However, to quantitatively estimate the uncertainties of the smoothed curves are difficult both because of the complexity of SDC500 processing steps and because there lacks quantitative indication of the uncertainty of MODIS surface reflectance product which is our main input. Although some of similar researches have adopted ablation strategy, i.e., to intentionally remove some observations and analysis the result, to evaluate a smoothing method, we found the ablation strategy not very pertinent to our problem because we actually can't ignore the uncertainty in the raw MODIS observations. This is especially true when the BRDF effect and undetected cloud exist in the raw MODIS observations. In response to the reviewer, we have rewritten section 5.4, added an analysis about the QA (quality assessment) and more discussions on the possible uncertainties in the SDC500 dataset. (p.34, lines 520-563)

**Reviewer #2**

**Comment 1: Seamless data of surface reflectance is required by various applications. This paper presented a study to generate 500m daily surface reflectance from MODIS data. The algorithm and assessment results were clearly documented. For further enhancement and clarity, I suggest considering the following recommendations:**

Response: Thank you for the positive opinion and constructive comments. They really helped us to improve the manuscript. We will respond your comments item by item.

**Comment 2: The defined reference geometry, characterized by a nadir view and a solar angle at 10:30 am, may not universally meet all application demands. Offering BRDF kernel parameters would be advantageous, facilitating users to adapt the data to their desired geometry.**

Response:

Thank you for the suggestion. We understand the strictness and capability of land surface BRDF. However, the purpose of SDC500 is to support remote sensing applications for public (non-professional) users, especially for the purpose to be combined with Landsat image. So, rather than building the dataset with BRDF parameters, such as the MODIS MCD43 series of products, we choose to build the SDC500 dataset with reflectance, which is more familiar to the public users.

**Comment 3: Many efforts were devoted to filling gaps in satellite data. The authors proposed a new approach combining math and ecosystem curve fitting. It would be beneficial if the rationale behind selecting this method were expanded upon. How does this newly introduced approach measure up against its pre-existing counterparts?**

Response:

Thank you for the professional comment. The idea of combining mathematical curves and ecosystem curve in fitting time series is a new attempt in reconstructing remote sensing time series. The ecosystem curve is an irregular curve which has only 2 parameters, i.e., scale and offset, but is representative to the typical change pattern of a land cover. Using ecosystem curve for a preliminary gap filling can provide a constraint to the later spline fitting, so that the spline fitting will not go off track even when there exists prolonged gap. And the mathematical curves (B-spline curves) have more flexible fitting ability, and can simulate the change of surface in a more realistic way. So, the combined use of ecosystem curve and mathematical curves, as described in section 3.3, can achieve a good balance between accuracy and stability in the presence of prolonged gap (cloud) in observation time series.

The performance of the new proposed approach can be illustrated with the example in section 5.2. The manuscript only presented spatial maps and statistics, while the time series curves may provide better insight to the result of the newly introduced method against traditional linear interpolation or spline fitting. The following figures (Fig. S3, Fig. S4 and Fig. S5) compares the reconstructed reflectance time series generated from the proposed method to that from a commonly used strategy, i.e., first fill gaps with linear gap-filling, then apply the slide-window smoothing to the filled time series to generate the final result.

Fig. S2 and Fig. S3 correspond to Fig. 20a,b and Fig. 20c,d in the main manuscript, respectively. We can see in Fig. S2 that the performance of the newly proposed SDC500 processing method is satisfactory, and the result of reduced input (Fig. S2b) is similar to result of full input (Fig. S2a), indicating the stability of SDC500 method in the extreme situation of prolonged gap around Julian day 180~260 (June and July). Fig. S3 is the reconstruction result of "linear gap-filling and smoothing", which is very commonly adopted in time series reconstruction. In Fig. S3, the reconstructed series (black solid curve) is not satisfactory, due to the influence of the noisy samples around Julian day 180~260; besides, when the good samples between days 195 to 199 are excluded from input (Fig. S3b), the final result changed significantly compared to the case of full sample (Fig. S3a), indicating the method is not stable when key sample points are missing.

Fig. S4 presents the reconstruction result of several more mathematical tools, with the same input samples as in Fig. S2b. We can see that because clear observation sample around Julian day 180~260 are scarce and noisy, the differences in reconstruction results are evident in this date range. The reconstructed line of double logistic curve fitting (blue solid line, Beck et al. 2006) lacks change details; the S-G filter (Savitzky & Golay, 1964; Chen et al., 2004) result with window of 80-days (orange solid line) looks good, but 80-days of filter window is too large, while the filter window is smaller than the gap length, i.e., window of 60-days (gray solid line) or smaller, the fitting result goes unreasonable; the DCT-PLS (penalized least-square regression based on discrete cosine transform) (Garcia, 2010; Yang et al., 2022) method is also proposed for time series reconstruction, but, its result (black solid line) gives false modulating signal around Julian day 180~260; in comparison, the result of our proposed SDC500 method is the most reasonable one in this example.

[Figure]

**Figure S3**   The performance of SDC500 gap-filling and smoothing algorithm in a pixel in South Asia, 2019. (a) with all the available MODIS observations as input; (b) with MODIS observations between days 195 to 199 excluded from input.

[Figure]

**Figure S4** The performance of linear gap-filling and smoothing algorithm in a pixel in South Asia, 2019. (a) with all the available MODIS observations as input; (b) with MODIS observations between days 195 to 199 excluded from input.

[Figure]

**Figure S5**  The performance of linear gap-filling and smoothing algorithm in a pixel in South Asia, 2019. (a) with all the available MODIS observations as input; (b) with MODIS observations between days 195 to 199 excluded from input.

Reference

Beck, P. S. A. , Atzberger, C. , Høgda, K. A. , Johansen, B. , & Skidmore, A. K. . (2006). Improved monitoring of vegetation dynamics at very high latitudes: a new method using MODIS NDVI. Remote Sensing of Environment, 99(3), 321-334.

Savitzky A., Golay M.J.E. Smoothing + Differentiation of Data by Simplified Least Squares Procedures. Analytical Chemistry 1964, 36, 1627

Chen, J., Jönsson, P., Tamura, M., Gu, Z., Matsushita, B., and Eklundh, L.: A simple method for reconstructing a high-quality NDVI time-series data set based on the Savitzky–Golay filter, Remote sensing of Environment, 91, 332-344, https://doi.org/10.1016/j.rse.2004.03.014, 2004.

Garcia D. Robust Smoothing of Gridded Data in One and Higher Dimensions with Missing Values. Comput Stat Data Anal 2010, 54, 1167-1178

K. Yang, Y. Luo, M. Li, S. Zhong, Q. Liu, and X. Li, "Reconstruction of Sentinel-2 Image Time Series Using Google Earth Engine," Remote Sens., vol. 14, no. 17, pp. 4395, Sep. 2022.

**Comment 5: An established seamless dataset exists: the MODIS nadir view BRDF-adjusted reflectance (NBAR) (Ju et al. 2010). Could the authors elucidate on the distinct advantages their new dataset presents over the gap-filled NBAR data?**

Response:

Thank you for providing the reference article. We are aware of the existence of a gap-filled NBAR in 30ArcSec spatial resolution (MCD43GF, in https://lpdaac.usgs.gov/products/mcd43gfv006/) (Sun et al. 2017), which is in lower resolution, and its generation was halted since 2018. However, the gap-filled NBAR in 500m spatial resolution, which is described in (Ju et al. 2010), is not a published dataset. The algorithms for NBAR gap-filling in both articles are important reference to

us. Out algorithm structure of outlier removal, temporal curve fitting and spatial filling are inherited from these former approaches. We have made some modifications in the introduction parts of the manuscript and added the reference article.

The difference between our SDC500 and gap-filled NBAR dataset are in several respects: 1) there is no available 500m gap-filled NBAR dataset; 2) SDC500 is based on daily surface reflectance while NBAR is based on BRDF parameters which are retrieved from 16-day composite of surface reflectance, so the gap-filled NBAR has longer process chain than that of SDC500, which is not the intention of this approach; 3) SDC500 is smoothed while the gap-filled NBAR is not smoothed, so, SDC500 looks more continuous than the gap-filled NBAR.

The following Figure S6 present a visual comparison of the available MCD43GF product vs our SDC500 dataset. In this comparison, SDC500 has been reprojected and upscaled to match the geometry of MCD43GF product. We can see from Figure S6 that there exist dark flecks in the upper part of MCD43GFimage, which is not shown in the SDC500 image. These dark flecks are false signals caused by unstable inversion of BRDF parameters in presence of heavy cloud in central Africa region. SDC500 dataset has removed most of the false signals during the outlier detection and smoothing processes, so the SDC500 image looks more continuous.

[Figure]

(a) MCD43GF                    (b) SDC500

**Figure S6** A illustration of the gap-filled image in central Africa, Julian day 220, 2017. (a) and (b) are from the MCD43GF (also called NBAR) product and our SDC500 product respectively. Color composite: R: band1, G: band2, B: band3.

**Comment 6: The construction of BRDF prior data employs a 100km x 100km grid. How does the values change across the grid boundary? Spatial filtering is suggested to smooth the boundary.**
Response: Thank you for the suggestion. The spatial interpolation or filtering is a way to solve the issue of across-boundary discontinuity. However, we adopted a different strategy to dealt with the issue in this approach. We have made an oversample to the BRDF prior, i.e., the granule for derive

the BRDF prior is 100km x 100km, but the spacing is only 50km. So, the cover areas of neighboring BRDF prior are overlapped. In this way, the issue of across- boundary discontinuity is mitigated. Furthermore, the main purpose of using BRDF prior is to ensure the stability of the process, while the most part of information comes from observation data, so the influence of BRDF prior to the final results is very small. Our visual inspections have found no across- boundary discontinuity so far. So, we guess that our strategy of oversample is enough at the current stage.

**Comment 7: The mixed land cover types might inadvertently introduce errors during LC-based BRDF normalization. How have such potential inconsistencies been addressed in this study?**

Response: It is a very good question. However, the mixed land cover within a pixel is a very challenging problem even for academic researches. And we are not intended to solve this problem, at least in this version of SDC500 product. As SDC500 is a long-term global dataset which involves large amount of computer resource and time to generate, some of the challenges can be left to the future as improvements in the next version.

We have added some discussion about mixed pixel, as well as other un-attended problem, such as the topographic effect, into section 5.4. (p.34, lines 520-563)

**Comment 8: A salient feature of the study is the distinct processing of snow-laden and snow-devoid seasons. This raises questions about other potential abrupt shifts, such as those induced by land cover alterations. What magnitude of uncertainty is attributed to such scenarios?**

Response: Thank you for raising this topic. It is desirable for the reconstructed dataset to truly reflect the abrupt change of land surface status, such scenarios usually include snow, fire, flood, etc. However, to balance between the ability to reflect the abrupt change and ability to resist noise or false signal is the most challenging problem in research of time series reconstruction. And our current research does not have breakthrough in this aspect. We have tried to keep an abrupt change between snow-laden and snow-free seasons, which can be observed in Fig. 5d, and Fig. 16a. We can't say that the performance is all satisfactory. For other abrupt changes, our method will smooth out the abrupt change to some extent.

Take the examples in section 5.4, Figure S7a and Figure S7b present the comparison between SDC500 and MCD43A4 (NBAR) product in the two corresponding sites of Fig. 22 and Fig. 23 in the manuscript. In the site near Anaktuvuk River, i.e., Figure S7a, the reported duration of tundra fire is about Julian day 240 to 285 in 2007, as indicated with gray bar in the figure. We can see from the "NBAR_2007" curve that there is an abrupt drop in the Normalized Burnt Ratio (NBR) index near day 250, while the "SDC_2007" curve shows a smoother drop, indicating that SDC500 has smooth-out effect to abrupt change. The rise of NBR near day 275 is caused by snow fall, the "SDC_2007" curve can reflect abrupt change during the switch of the snow/snow-free status.    In the site of Australia forest fire, i.e., Figure S7b, the reported duration of fire is about Julian day 300 to 350 in 2019, as indicated with gray bar. We can see that there is a gap in the "NBAR_2019" curve, which is probably caused by the heavy smoke of the forest fire. The the "SDC_2019" curve, is continuous and smooth, just as expected.

We have modified section 5.3 and add some discussion about abrupt change in the revised manuscript. (p.33, lines 510-513)

[Figure]

**Figure S7** Comparison between SDC500 and MCD43A4 (NBAR) product in two sites of wild fire disturbance. (a) site near Anaktuvuk River, 2007; (b) site of Australia forest fire, 2019.

**Comment 9: The data sets, in their current form, seem to lack integral quality control metrics, which would immensely aid end users . For example, the above-mentioned cases of abrupt changes (snow, land cover change) can be labelled for user convenience.**

Response: Thank you for the constructive comment. The SDC500 dataset have a band of quality assessment (QA) flag. We have added a description of the QA flag in section 5.4 (p.34, lines 520-563). The reviewer's suggestion to label the abrupt changes for user convenience is very good. We will add such an indication flag in the QA flag, in our next version of SDC500 dataset. Considering that As SDC500 is a long-term global dataset which involves large amount of computer resource and time to generate, this new flag cannot be added to the current version of dataset.

Reference
Junchang Ju, David P. Roy, Yanmin Shuai, Crystal Schaaf, Development of an approach for generation of temporally complete daily nadir MODIS reflectance time series, Remote Sensing of Environment, Volume 114, Issue 1, 2010, Pages 1-20

**Reviewer #3**

**Comment 1:**

**The paper proposes a BRDF-corrected gap-filled MODIS dataset. Overall the paper reads well and looks scientifically sound. I was able to download a few tiles of the dataset over Asia, but not over the test sites. I was especially interested into downloading those over the Amazon forests, where I am used to the reflectance values and its spatial patterns, and I would've been able to confirm if the data are making sense. Therefore, at this momment I am unable to confirm if data are good by my own inspection. Since this is a dataset paper, I cannot recommend its acceptance without looking at data for locations I can give an expert opinion. I apologize in advance if the data were available and I was not able to find them, please let me know and I'll be willing to look at it.**

Response:

Thank you for reviewing the manuscript. We are truly sorry for the inconvenience you've experienced in downloading the SDC500 dataset. We have previously encountered some technical problems in the data-sharing website because of the large number of data files, and restricted the downloadable content to only a few MODIS tiles. When we received you comments, we re-configured the data-sharing website. Now the data users can access all the global dataset, but need to go to different web pages to access dataset of different year. The re-configured website is still in: https://data-starcloud.pcl.ac.cn/resource/27, with anonymous user name "test1@pcl.ac.cn" and password "1234567s".

We welcome fellow researchers to download and use the SDC500 dataset, and very much appreciate evaluations, comments and suggestions to the dataset. The generation of global seamless data cube in multiple resolutions (500m, 250m, 30m, 10m) is be a long-term task for our team. Dataset in other resolutions will come to public soon. And the current version of SDC500 dataset will also be upgraded to overcome the existing flaws. So, comments from data users the most precious asset which will guide us in improving the datasets.

**Specific comment:**

**Figure 22, non-fired -> "unburned" would look better**

Response:

Thank you for the suggestion. We have modified it in the revised manuscript.